# LABEL-FREE CONCEPT BOTTLENECK MODELS

**Tuomas Oikarinen**
UCSD CSE
toikarinen@ucsd.edu

**Subhro Das**
MIT-IBM Watson AI Lab, IBM Research
subhro.das@ibm.com

**Lam M. Nguyen**
IBM Research
lamnguyen.mltd@ibm.com

**Tsui-Wei Weng**
UCSD HDSI
lweng@ucsd.edu

## ABSTRACT

Concept bottleneck models (CBM) are a popular way of creating more inter-pretable neural networks by having hidden layer neurons correspond to human-understandable concepts. However, existing CBMs and their variants have two crucial limitations: first, they need to collect labeled data for each of the predefined concepts, which is time consuming and labor intensive; second, the accuracy of a CBM is often significantly lower than that of a standard neural network, especially on more complex datasets. This poor performance creates a barrier for adopting CBMs in practical real world applications. Motivated by these challenges, we propose *Label-free* CBM which is a novel framework to transform any neural network into an interpretable CBM without labeled concept data, while retaining a high accuracy. Our Label-free CBM has many advantages, it is: *scalable* - we present the first CBM scaled to ImageNet, *efficient* - creating a CBM takes only a few hours even for very large datasets, and *automated* - training it for a new dataset requires minimal human effort. Our code is available at https://github.com/Trustworthy-ML-Lab/Label-free-CBM.

## 1 INTRODUCTION

Deep neural networks (DNNs) have demonstrated unprecedented success in a wide range of machine learning tasks such as computer vision, natural language processing, and speech recognition. However, due to their complex and deep structures, they are often regarded as *black-box* models that are difficult to understand and interpret. Interpretable models are important for many reasons such as creating calibrated trust in models, which means understanding when we should trust the models. Making deep learning models more interpretable is an active yet challenging research topic.

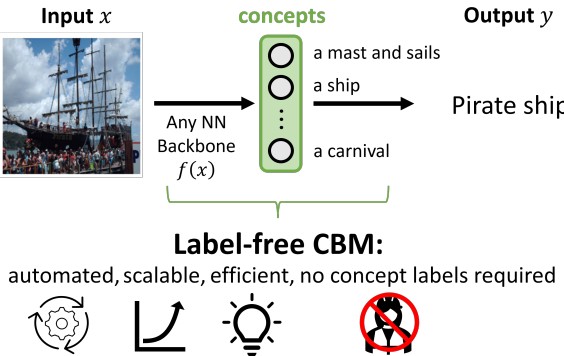

Figure 1: Our proposed Label-free CBM has many desired features which existing CBMs lack, and it can transform any neural network backbone into an interpretable Concept Bottleneck Model.

One approach to make deep learning more interpretable is through Concept Bottleneck Models (CBMs) (Koh et al., 2020). CBMs typically have a Concept Bottleneck Layer before the (last) fully connected layer of the neural network. The concept bottleneck layer is trained to have each neuron correspond to a single human understandable *concept*. This makes the final decision a linear function of interpretable concepts, greatly increasing our understanding of the decision making. Importantly, CBMs have been shown to be useful in a variety of applications, including model debugging and human intervention on decisions. However, there are two crucial limitations of existing CBMs and their variants (Koh et al., 2020; Yuksekgonul et al., 2022; Zhou et al., 2018): (i) labeled data is required for each of the predefined concepts, which is time consuming and expensive to collect; (ii) the accuracy of a CBM is often significantly lower than the accuracy of a standard neural network, especially on more complex datasets.

To address the above two challenges, we propose a new framework named Label-free CBM, which is capable of transforming any neural network into an interpretable CBM *without* labeled concept data while *preserving* accuracy comparable to the original neural network by leveraging foundation models (Bommasani et al., 2021). Our Label-free CBM has many advantages:

- it is *scalable* - to our best knowledge, it is the first CBM that scales to ImageNet
- it is *efficient* - creating a CBM takes only a few hours even for very large datasets
- it is *automated* - training it for a new dataset requires minimal human effort

## 2 RELATED WORK

**Post-hoc explanations** (Samek et al., 2021): The approach of post-hoc explanations includes some classic methods such as LIME (Ribeiro et al., 2016) and SHAP (Lundberg & Lee, 2017), which try to explain individual model decisions by identifying which parts of the input data (e.g. pixels) are the most important for a given decision. However, these methods are based on local approximations of the DNN model and as such are not always accurate. Further, the explanations at the granularity of input pixels may not always be helpful and could require substantial subjective analysis from human. In contrast, our explanations in Section 4 are not approximated and explain predictions in terms of human-understandable concepts.

**More interpretable final layer**: (Wong et al., 2021) proposes making the FC layer sparse, and develop an efficient algorithm for doing so. They show that sparse models are more interpretable in many ways, but it still suffers from the fact the previous layer features are not interpretable. NBDT (Wan et al., 2020) propose replacing the final layer with a neural backed decision tree for another form of more interpretable decisions. Other approaches to make NNs more interpretable include Concept Whitening (Chen et al., 2020) and Concept Embedding Models (Zarlenga et al., 2022).

**CBM**: Most related to our approach are Concept Bottleneck Models (Koh et al., 2020; Losch et al., 2019) which create a layer before the last fully connected layer where each neuron corresponds to a human interpretable concept. CBMs have been shown to be beneficial by allowing for human test-time intervention for improved accuracy, as well as being easier to debug. To reduce the training cost of a CBM, a recent work (Yuksekgonul et al., 2022) proposed Post-Hoc CBM that only needs to train the last FC layer along with an optional residual fitting layer, avoiding the need to train the backbone from scratch. This is done by leveraging Concept Activation Vectors (CAV) (Kim et al., 2018) or the multi-modal CLIP model (Radford et al., 2021). However, the post-hoc CBM does not fully address the problems of the original CBM as using TCAV still requires collecting annotated concept data and their use of CLIP model can only be applied to if the NN backbone is the CLIP image encoder. Additionally, the performance of post-hoc CBMs without uninterpretable residual fitting layers is often significantly lower than the standard DNNs. Similarly, an earlier work Interpretable Basis Decomposition (Zhou et al., 2018) proposes learning a concept bottleneck layer based on labeled concept data for explanable decisions, even though they do not call themselves a CBM. Comparison between the features our method and existing approaches is shown in Table 1.

**Model editing/debugging**: Our approach is related to a range of works proposing ways to edit networks, such as (Bau et al., 2020; Wang et al., 2022) for generative vision models, (Bau et al., 2020) for classifiers, or (Meng et al., 2022; Mitchell et al., 2021) for language models. In addition (Abid et al., 2021) propose a way to debug model mistakes using TCAV activation vectors.

|  | | (I) Flexibility | | (II) Interpretability | | (III) Performance | |
|---|---|---|---|---|---|---|---|
| Method: | | Without labeled concept data | Any network architecture | Sparse final layer | All features interpretable | Preserves accuracy | Extends to ImageNet scale |
| CBM | | No | **Yes** | No | **Yes** | No | No |
| IBD | | No | **Yes** | No | No | **Yes** | No |
| P-CBM | | No | **Yes** | Yes | **Yes** | No | No |
| P-CBM (CLIP) | | **Yes** | No | **Yes** | **Yes** | No | Maybe |
| P-CBM-h | | No | **Yes** | **Yes** | No | **Yes** | No |
| P-CBM-h (CLIP) | | **Yes** | No | **Yes** | No | **Yes** | Maybe |
| Label-free CBM **(This work)** | | **Yes** | **Yes** | **Yes** | **Yes** | **Yes** | **Yes** |

Table 1: Comparison of our method against existing methods for creating Concept Bottleneck models, CBM (Koh et al., 2020), IBD (Zhou et al., 2018) and 4 versions of P-CBM (Yuksekgonul et al., 2022), where '-h' indicates the hybrid model that uses uninterpretable residual term, '(CLIP)' means models using CLIP concepts. We used maybe to indicate models that could in theory extend to ImageNet but have not been tested.

**CLIP-Dissect** (Oikarinen & Weng, 2022): CLIP-Dissect is a recent method for understanding the roles of hidden layer neurons by leveraging the CLIP multimodal model (Radford et al., 2021). It can provide a score of how close any neuron is to representing any given concept without the need of concept annotation data, which makes it useful as an optimization target for learning our interpretable projection in Step 3.

## 3 LABEL-FREE CBM: A NEW FRAMEWORK TO BUILD CBM

In this section, we propose Label-free CBM, a novel framework that builds a concept bottleneck model (CBM) in an *automated*, *scalable* and *efficient* fashion and addresses the core limitations of existing CBMs. Given a neural network backbone, Label-free CBM transforms the backbone into an interpretable CBM without the need of concept labels with the following 4 steps which are illustrated in Fig 2 – **Step 1:** Create the initial concept set and filter undesired concepts; **Step 2:** Compute embeddings from the backbone and the concept matrix on the training dataset; **Step 3:** Learn projection weights $W_c$ to create a Concept Bottleneck Layer (CBL); **Step 4:** Learn the weights $W_F$ of the sparse final layer to make predictions.

Note that the backbone model can either be a model trained on the target task, or a general model trained on a different task. The details of Label-free CBM for Step 1 are provided in Sec 3.1, Step 2 and Step 3 in Sec 3.2, and finally Step 4 in Sec 3.3.

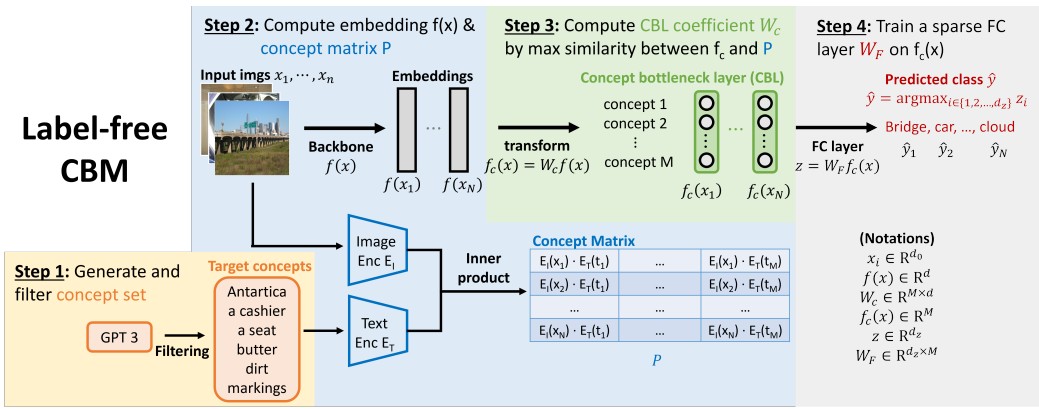

Figure 2: Overview of our pipeline for creating label-free CBM.

### 3.1 STEP 1: CONCEPT SET CREATION AND FILTERING

In this step, we will describe how to create a concept set to serve as the basis of human-interpretable concepts in the Concept Bottleneck Layer. This step consists of two sub-steps: **A. Initial concept set creation** and **B. Concept set filtering**.

**A. Initial concept set creation**: A concept set refers to the set of concepts represented in the Concept Bottleneck Layer. In the original CBM paper (Koh et al., 2020), this is decided by domain experts as the set of concepts that are important for the given task. However, since our objective is to automate the entire process of generating CBMs, we don't want to rely on human experts. Instead, we propose generating the concept set via GPT-3 (Brown et al., 2020) using the OpenAI API. Somewhat surprisingly, GPT-3 has a good amount of domain knowledge of which concepts are important for detecting each class when prompted in the right way. Specifically, we ask GPT-3 the following:

- List the most important features for recognizing something as a {class}:
- List the things most commonly seen around a {class}:
- Give superclasses for the word {class}:

Note that the {class} in here refers to the class name in the machine learning task. For GPT-3 to perform well on the above prompt, we provide two examples of the desired outputs for few-shot adaptation. Note that those two examples can be shared across all datasets, so no additional user input is needed for generating concept set for a new dataset. Full prompts and example outputs are illustrated in the Appendix Figures 10 and 11. To reduce variance we run each prompt twice and combine the results. Combining the concepts received from different classes and prompts gives us a large, somewhat noisy set of initial concepts, which we further improve by filtering. We found using GPT-3 to generate initial concepts to perform better than using the knowledge-graph ConceptNet (Speer & Havasi, 2013) which was used in Post Hoc-CBM (Yuksekgonul et al., 2022). Reasons for this and a comparison between the two are presented in Appendix A.6.

**B. Concept set filtering**: Next we employ several filters to improve the quality and reduce the size of our concept set, as stated below:

1. *Concept length*: We delete any concept longer than 30 characters in length, to keep concepts simple and avoid unnecessary complication.

2. *Remove concepts too similar to classes*: We don't want our CBM to contain the output classes themselves as that would defeat the purpose of an explanation. To avoid this we remove all concepts that are too similar to the names of target classes. We measure this with cosine similarity in a text embedding space. In particular we use an ensemble of similarities in the CLIP ViT-B/16 text encoder as well as the all-mpnet-base-v2 sentence encoder space, so our measure can be seen as a combination of visual and textual similarity. For all datasets, we deleted concepts with similarity $> 0.85$ to any target class.

3. *Remove concepts too similar to each other*: We also don't want duplicate or synonymous concepts in the bottleneck layer. We use the same embedding space as above, and remove any concept that has another concept with $> 0.9$ cosine similarity already in concept set.

4. *Remove concepts not present in training data*: To make sure our concept layer accurately presents its target concepts, we remove any concepts that don't activate CLIP highly. This cut-off is dataset specific, and we delete all concepts with average top-5 activation below the cut-off.

5. *Remove concepts we can't project accurately*: Remove neurons that are not interpretable from the CBL. This step is actually performed after step 3 and is described in section 3.2.

We discuss the reasoning behind the filters in detail in Appendix A.5 and perform an abblation study on their effects in Appendix A.4.

### 3.2 STEP 2 AND 3: LEARNING THE CONCEPT BOTTLENECK LAYER (CBL)

Once the concept set is obtained, the next step is to learn a projection from the backbone model's feature space into a space where axis directions correspond to interpretable concepts. Here, we

present a way of learning the projection weights $W_c$ without any labeled concept data by utilizing CLIP-Dissect (Oikarinen & Weng, 2022). To start with, we need a set of target concepts that the bottleneck layer is expected to represent as $\mathcal{C} = \{t_1, ..., t_M\}$, as well as a training dataset (e.g. images) $\mathcal{D} = \{x_1, ..., x_N\}$ of the original task. Next we calculate and save the CLIP concept activation matrix $P$ where $P_{i,j} = E_I(x_i) \cdot E_T(t_j)$ and $E_I$ and $E_T$ are the CLIP image and text encoders respectively. $W_c$ is initialized as a random $M \times d_0$ matrix where $d_0$ is the dimensionality of backbone features $f(x)$. The initial set $\mathcal{C}$ is created in Step 1 and the training set $\mathcal{D}$ is provided by the downstream task. We define $f_c(x) = W_c f(x)$, where $f_c(x_i) \in \mathbb{R}^M$. We use $k$ to denote a neuron of interest in the projection layer, and its activation pattern is denoted as $q_k$ where $q_k = [f_{c,k}(x_1), \ldots, f_{c,k}(x_N)]^\top$, with $q_k \in \mathbb{R}^N$ and $f_{c,k}(x) = [f_c(x)]_k$.

To make the neurons in the CBL interpretable, we need to enforce the projected neurons to activate in correlation with the target concept, which we do by optimizing $W_c$ to maximize the CLIP-Dissect similarity between the neuron's activation pattern and the target concept. To optimize this similarity, we have designed a new fully differentiable similarity function $\text{sim}(t_i, q_i)$ that can be applied to CLIP-Dissect, called *cos cubed*, that still achieves very good performance in explaining the neuron functionality as shown in Appendix A.3. Our optimization goal is to minimize the objective $L$ over $W_c$ as defined in Equation (1):

$$L(W_c) = \sum_{i=1}^{M} -\text{sim}(t_i, q_i) := \sum_{i=1}^{M} -\frac{\bar{q}_i^3 \cdot \bar{P}_{:,i}^3}{||\bar{q}_i^3||_2 ||\bar{P}_{:,i}^3||_2}. \tag{1}$$

Here $\bar{q}$ indicates vector $q$ normalized to have mean 0 and standard deviation 1, and the *cos cubed* similarity $\text{sim}(t_i, q_i)$ is simply the cosine similarity between two activation vectors after both have been normalized and raised to third power element-wise. The third power is necessary to make the similarity more sensitive to highly activating inputs. As this is still a cosine similarity, it takes values between $[-1, 1]$. We optimize $L(W_c)$ using the Adam optimizer on training data $\mathcal{D}$, with early stopping when similarity on validation data starts to decrease. Finally to make sure our concepts are truthful, we drop all concepts $j$ with $\text{sim}(t_j, q_j) < 0.45$ on validation data after training $W_c$. This is the 5th concept set filter from Sec 3.1. This cutoff was selected manually as a good indicator of a neuron being interpretable. During this filtering, the number of concept is reduced: $M \leftarrow M - \Delta$, where $\Delta$ is non-negative integer representing the number of concepts being removed in this step. Note that matrix $W_c$ has also to be updated accordingly by removing the rows that corresponds to the removed concepts, and with our notation, $W_c \in \mathbb{R}^{M \times d_0}$. To simplify Figure 2, we omit plotting this concept removal step.

## 3.3 STEP 4: LEARNING THE SPARSE FINAL LAYER

Now that the Concept Bottleneck Layer is learned, the next task is to learn the final predictor with the fully connected layer $W_F \in \mathbb{R}^{d_z \times M}$ where $d_z$ is the number of output classes. The goal is to keep $W_F$ sparse, since sparse layers have been demonstrated to be more interpretable (Wong et al., 2021). Given that both the backbone $f(x)$ and learned concept projection $W_c$ are fixed, this is a problem of learning a sparse linear model, which can be solved efficiently with the elastic net objective:

$$\min_{W_F, b_F} \sum_{i=1}^{N} L_{ce}(W_F f_c(x_i) + b_F, y_i) + \lambda R_\alpha(W_F) \tag{2}$$

where $R_\alpha(W_F) = (1 - \alpha)\frac{1}{2}||W_F||_F + \alpha||W_F||_{1,1}$, $|| \cdot ||_F$ denotes the Frobenius norm, $|| \cdot ||_{1,1}$ denotes element wise matrix norm, $b_F$ denotes the bias of the FC layer, $L_{ce}$ is the standard cross-entropy loss and $y_i$ is the ground-truth label of data $x_i$. We optimize Equation (2) using the GLM-SAGA solver created by (Wong et al., 2021). For the sparse models, we used $\alpha = 0.99$ and $\lambda$ was chosen such that each model has 25 to 35 nonzero weights per output class. This level was found to still result in interpretable decisions while retaining good accuracy. Depending on the number of features/concepts in the previous layer this corresponds to 0.7-15% of the weights of the model being nonzero.

## 4    EXPERIMENT RESULTS

We present three main results on evaluating the accuracy and interpretability of the Label-free CBM in this section. Due to page limit, additional experiments and discussions are in Appendix A.2-A.12. An overview of additional experiments is provided in Appendix A.1.

**Datasets.** To evaluate our approach, we train Label-free CBMs on 5 datasets. These are CIFAR-10, CIFAR-100 (Krizhevsky et al., 2009), CUB (Wah et al., 2011), Places365 (Zhou et al., 2017) and ImageNet (Deng et al., 2009). This is a diverse set of tasks, where CIFAR-10/100 and ImageNet are general image classification datasets, CUB is a fine-grained bird-species classification dataset and Places365 is focused on scene recognition. Their sizes vary greatly, with CUB having 5900 training samples, CIFAR datasets 50,000 each and ImageNet and Places365 have 1-2 million training images. CUB contains annotations with 312 concepts for each datapoint, such as *has wing color:blue* or *has head pattern:spotted*. We used neither the concept names or the concept annotations to train our LF-CBM models to showcase our ability to perform without labels, yet our method discovered similar concepts and is competitive with methods that utilize the available concept information.

**Setup.** For CIFAR and CUB we use the same backbone models as (Yuksekgonul et al., 2022) for fair comparison, so we use CLIP(RN50) image encoder as the backbone for CIFAR, and ResNet-18 trained on CUB from imgclsmob for CUB. For both ImageNet and Places365 we use ResNet-50 trained on ImageNet as the backbone network. The number of concepts each model uses is roughly proportional to the number of output classes for that task, as each class adds more initial concepts. The number of concepts for our models are as follows: 128 for CIFAR-10, 824 for CIFAR-100, 211 for CUB, 2202 for Places-365 and 4505 for ImageNet. CUB has a smaller number of concepts because we only used the *important features* prompt. All models are trained on a single Nvidia Tesla P100 GPU, and the full training run takes anywhere from few minutes to 20 hours depending on the dataset size. The majority of runtime is taken by step 2, where we save activations for both the backbone and CLIP over the entire training dataset. Fortunately, these results only need to be calculated once and can be reused for training new CBMs on the same dataset. Once the activations have been saved, learning the model takes less than 4 hours on all datasets.

**Result (I): Accuracy.** Table 2 shows the performance of Label-free CBM on all 5 datasets. We can see that our method can create a CBM with a sparse final layer on with little loss in accuracy on all datasets, including a model with 72% top-1 accuracy on the ImageNet. Our Label-free CBM has significantly higher accuracy than Post-hoc CBM on the datasets we evaluated, but some rows for P-CBM are N/A as they do not provide results those results and it is unclear how to scale P-CBM to larger datasets. For Table 2 we excluded methods with non-interpretable components (i.e. P-CBM-h or IBD) as we want to focus on fully interpretable CBM models. Note that P-CBM uses the expert provided concept set for CUB200, yet we can outperform it using our fully GPT-3 derived set of features. The standard sparse models were finetuned by us by learning a sparse final layer directly after feature layer $f(x)$ as described in (Wong et al., 2021) and also have 25-35 nonzero weights per class. The accuracies of sparse standard models are comparable to our CBM, indicating the CBM does not reduce accuracy.

| | | Dataset | | | | |
|---|---|---|---|---|---|---|
| Model | Sparse final layer | CIFAR10 | CIFAR100 | CUB200 | Places365 | ImageNet |
| Standard | No | 88.80% | 70.10% | 76.70% | 48.56% | 76.13% |
| Standard (sparse) | Yes | 82.96% | 58.34% | **75.96%** | 38.46% | **74.35%** |
| P-CBM* | Yes | 70.50% | 43.20% | 59.60% | N/A | N/A |
| P-CBM (CLIP)* | Yes | 84.50% | 56.00% | N/A | N/A | N/A |
| Label-free CBM **(Ours)** | Yes | **86.40%** ± 0.06% | **65.13%** ± 0.12% | 74.31% ± 0.29% | **43.68%** ± 0.10% | 71.95% ± 0.05% |

Table 2: Accuracy comparison, best performing sparse model bolded. We can see our method outperforms Posthoc-CBM and performs similarly to a sparse standard model. The results for our method are mean and standard deviation over three training runs. *Indicates reported accuracy.

**Result (II): Explainable decision rules.** Perhaps the biggest benefit of Concept Bottleneck Models is that their decisions can be explained as a simple linear combination of understandable features. To showcase this, in Figure 3 we visualize the final layer weights for the classes "Orange" and "Lemon"

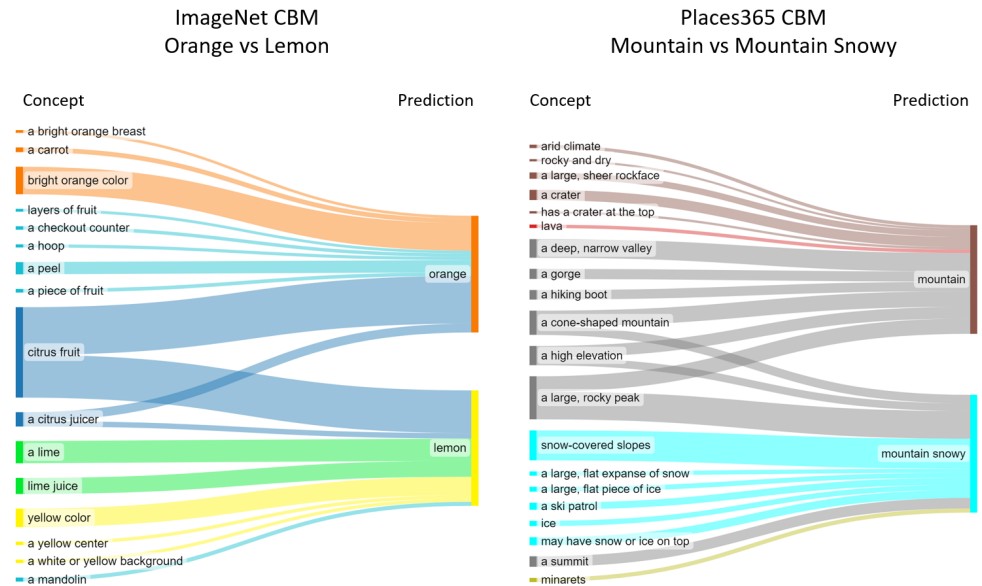

Figure 3: Visualization of the final layer weights of our Label-free CBM. Showcasing how our models differentiate between two similar classes.

on ImageNet, and classes "Mountain" and "Mountain Snowy" on Places365. This visualization is a Sankey diagram of the final layer weights coming into two output classes, with the weight between a concept and output class displayed as the width of the line connecting them. We have only included weights with absolute value greater than 0.05 (for comparison the largest weights are usually 0.5-1). Negative weights are denoted as NOT concept. The learned decision rules largely align with our intuitions. For ImageNet, the concept "citrus" fruit is highly connected to both "Orange" and "Lemon", while orange colors activate the "Orange" class and yellow colors and limes activate the "Lemon" class. On Places, concepts like "a high elevation" activate both classes, while "Mountain Snowy" is activated by snow and ice related concepts and "Mountain" is activated by concepts related to volcanoes like "lava" and "crater". Visualizations like this allows us to gain a *global* understanding of how our model behaves. Additional visualizations are shown in Appendix A.11

**Result (III): Explainable individual decisions.** In addition to global understanding, CBMs allow us to understand the reasoning behind individual decisions. Since our decisions are linear functions of interpretable features, they can be explained in a simple and accurate way. After learning the Concept Bottleneck Layer, we normalize the activations of each concept to have mean 0 and standard deviation 1 on the training data. Given normalized features, the contribution of feature $j$ to output $i$ on input $x_k$ can be naturally calculated as $Contrib(x_k, i, j) = W_{F[ij]} f_c(x_k)_j$. Since our $W_F$ is sparse, most contributions will be 0 and the important contributions can be easily visualized with a bar plot. Example visualizations for explaining the models predicted class are shown in Figure 4, and more visualizations are available in Appendix A.12. These visualizations show the concepts with highest absolute value contribution. Concepts with negative activation can still be important contributors, and they are shown as "NOT concept" in our visualizations. We can see for example the CUB model correctly identifies "a red head" as the most important feature in recognizing this image as "Red headed Woodpecker", while the fact that there is no "a rosy breast" present also increases models confidence.

## 5 CASE STUDY: MANUALLY IMPROVING AN IMAGENET MODEL

In this section we take a deep dive into our ImageNet Label-free CBM – we analyze the errors it makes and design ways to debug/improve our network based on these findings. In particular, we show how we can inspect a handful of individual incorrect decisions and manually change our model weights to not only fix those predictions but also multiple other predictions, improving the networks

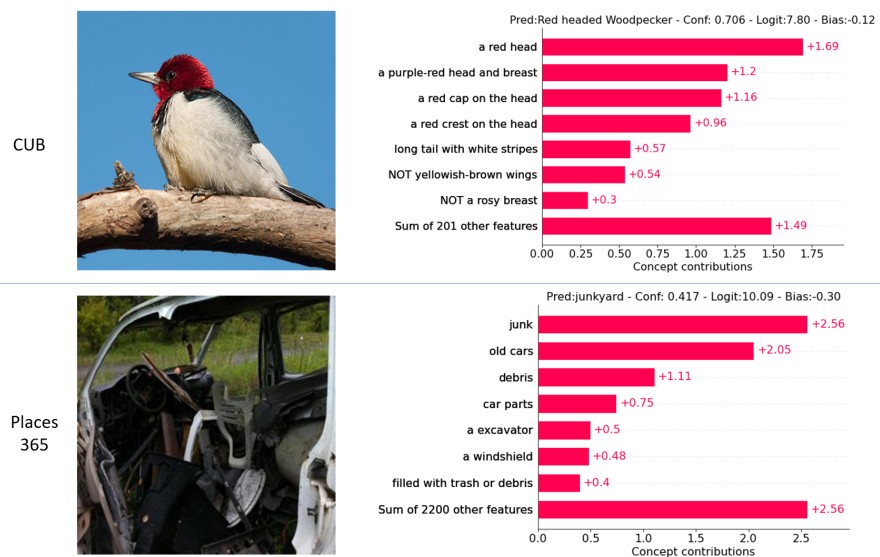

Figure 4: Visualization of two correct decisions made by our Label-free CBM.

overall accuracy. To our knowledge, this is the first example of manually editing a large well-trained neural network in a way that improves test accuracy on the same distribution.

## 5.1 TYPES OF MODEL ERRORS

During the course of our experiment, we were able to identify 4 different types of mistakes our model makes, with different strategies for fixing them:

**Type 1: Incorrect/ambiguous label**: Despite looking like errors in terms of accuracy, these are simply a consequence of having noisy labels and are unavoidable. We will not attempt to fix this type of mistakes.

**Type 2: No sufficient concept in CBL**: Despite having a large set of concepts, our CBL does not include all concepts important for detecting certain classes. For example, detecting the difference between two species of snakes may require recognizing very fine-grained patterns that can't be easily explained in words. While these could often be fixed by adding new concepts to the concept set, we do not focus on it as it requires retraining the projection $W_c$ and the final layer.

**Type 3: Incorrect concept activations**: Some of the time, the activations in our CBL are incorrect and do not match the image, causing the prediction to become incorrect. Since the network up to the Concept Bottleneck Layer is mostly a black box, we can't easily improve the predictions, but predictions can be improved with test time interventions as shown in Figure 5.

**Type 4: Incorrect final layer weight** $W_F$: Sometimes even if all the concept activations are correct, the models final layer weights still cause it to make an erroneous prediction. As the final layer is fully interpretable, fixing these errors is the main focus of our study described in Sec 5.2.

Examples of Type 1 and Type 2 errors are shown in Figure 13 in the Appendix.

## 5.2 EDITING FINAL LAYER WEIGHTS

In this section we describe our procedure for manually editing final layer weights of our model. This is done with the following procedure:

1. **Find an input where the model makes a Type 4 error**: This is done by visualizing incorrect model predictions and their explanations, and identifying Type 4 errors, i.e. errors where the highly activating concepts look correct and the label is correct and unambiguous.

2. **Identify a concept to edit.**: Perhaps the most tricky step, to change a prediction we need to select a concept that is highly activated in this image, important for the ground truth class and not important for the incorrect (using our domain knowledge), and will have a small impact on other

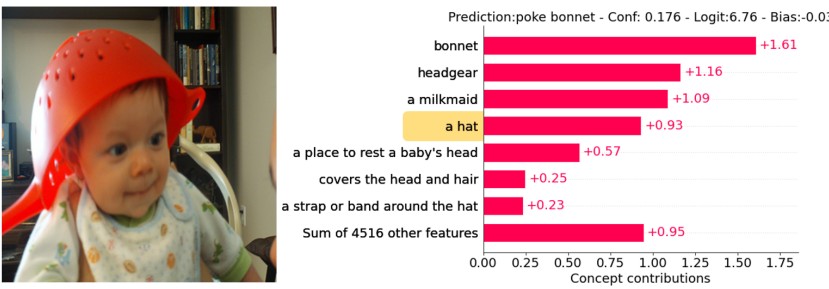

Figure 5: Here we fix an incorrect model prediction on ImageNet by simply zeroing out the incorrect activation for concept "a hat". Model originally predicted "poke bonnet" which is a type of hat.

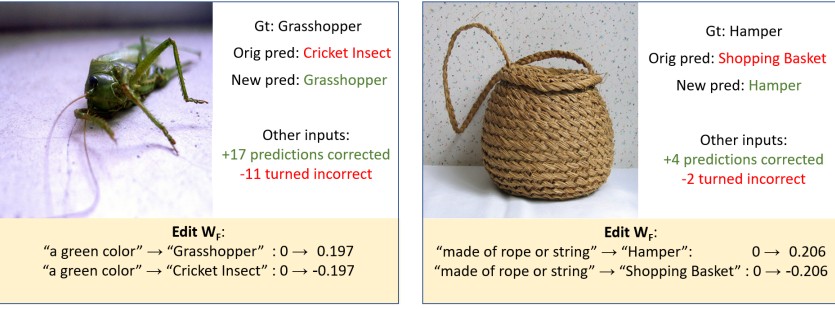

Figure 6: Examples of two edits we performed on our model predictions. Both edits fix the error we found, and fix more other predictions than they cause mistakes on other unseen inputs.

predictions. For example to flip a prediction from "cricket insect" to "grasshopper" we chose the concept "a green color".

3. **Change weights by sufficient magnitude**: Next we choose magnitude of weight change $\Delta w$, and edit the weights in the following way: $W_{F[gt,concept]} \leftarrow W_{F[gt,concept]} + \Delta w$ and $W_{F[pred,concept]} \leftarrow W_{F[pred,concept]} - \Delta w$. Typically we want $\Delta w$ to be slightly more than required to flip the prediction on our example, but understanding its effect on other predictions is hard and should be checked with a validation dataset.

Fig 6 shows two examples of model edits we performed, and their effects on model predictions on the validation set. In total we identified 5 such beneficial edits during our quick exploration, the other 3 are shown in Fig 12. After applying all of them, the total validation accuracy of our model goes from 71.98% to 72.02%, correcting 38 predictions while turning 17 incorrect. While this might seem like a small difference, it is worth noting we only change 10/4.5 million total weights in the final layer, of which 34,550 are non-zero. Our edits only affect 10 classes which have a total of 500 validation examples, so in effect we increase accuracy on this subset by 4.2% which is a significant boost. We believe this editing approach is a promising direction for future use, where practitioners could for example notice an incorrect prediction in a production system, quickly devise a fix for that prediction, and check how it affects other predictions using validation data.

## 6 CONCLUSION

We have presented Label-free CBM, a fully automated and scalable method for generating concept bottleneck models and used it to create the first high performing CBM on ImageNet. In addition, we have demonstrated how our models are easier to understand in terms of both global decision rules and reasoning for individual predictions. Finally, we show how to use this understanding to manually edit weights to improve accuracy in a trained ImageNet model.

ACKNOWLEDGEMENTS

The authors would like to thank anonymous reviewers for valuable feedback to improve the manuscript. The authors also thank MIT-IBM Watson AI lab for support in this work. This work was done during T. Oikarinen's internship at MIT-IBM Watson AI Lab. T.-W. Weng is supported by National Science Foundation under Grant No. 2107189.

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

# A APPENDIX

## A.1 APPENDIX OVERVIEW

In this section we provide a brief overview of the Appendix contents. The Appendix is mostly focused on Ablation studies identifying the importance and effectiveness of different components of our pipeline, as well additional Figures to showcase more examples of our results.

First in section A.2 we discuss the limitations of our method, then in section A.3 we show that our proposed *cos cubed* similarity function performs well in describing neuron functionality. In section A.4 we perform an ablation study on the effect of different concept set filters we have proposed, and in Sec A.5 we discuss our reasoning behind each filter in detail through an example. In section A.6 we show that our GPT-3 generated concept set helps us reach higher accuracy than previous methods for creating initial concept set. In Appendix A.7 we show the effects of removing the sparsity constraint on our models, leading to higher accuracy but reduced interpretability. In Appendix A.8 we show our results are consistent despite inherent randomness in our pipeline. Next, in Appendix A.9 we provide a detailed explanation of the procedure we used for model editing described in Sec 5.2. Finally in Sections A.10, A.11 and A.12 we provide additional figures for model editing, individual decision explanations and model weight visualizations respectively.

## A.2 LIMITATIONS

While our model performs well in terms of both interpretability and accuracy, it still has some limitations. For example, using a model like GPT-3 to produce concepts is stochastic, and sometimes may fail to generate important concepts for detecting a certain class. However our model is quite robust to small changes in the concept set as shown in sections A.4 and A.8. In general, automatic ways of generating concept set may sometimes lack the required domain knowledge to include some important concepts, and would perhaps be best used in collaboration with a human expert.

Second, our model is aimed to extend Concept Bottleneck Models to large datasets where labeled concept data is not available. However it works best on domains where CLIP performs well and as such may not perform as well on small datasets that require specific domain knowledge and labeled concept data is available, such as medical datasets. For such datasets we believe it is still better to use a method that can effectively leverage the labels available such as original Concept Bottleneck Models Koh et al. (2020) or P-CBMYuksekgonul et al. (2022).

## A.3 DISSECTION PERFORMANCE OF COS CUBED SIMILARITY FUNCTION

In this section we discuss how good the *cos cubed* similarity function we defined in section 3.2 is at describing neuron functionality, i.e. how well it performs as a similarity function for CLIP-Dissect. Specifically we measured its performance on describing the roles of final layer neurons on a ResNet-50 trained on ImageNet (where we know the ground truth role of the neurons) in terms of both accuracy and average distance from the correct description in a text embedding space. We followed the quantitative evaluation methodology of (Oikarinen & Weng, 2022), which describes this process in more detail. We compared the results against using CLIP-Dissect with two different similarity functions, *softWPMI* which was the best function identified by (Oikarinen & Weng, 2022) and *cos* which is simple a differentiable baseline. We can see *cos cubed* performs es well as *softWPMI* in terms of similarity in text embedding space, while being slightly worse in terms of accuracy, and is much better than simple *cos* similarity. This is impressive given *cos cubed* is fully differentiable while *SoftWPMI* is not differentiable.

| Metric | Similarity function | $D_{probe}$ | | | | | |
| | | CIFAR100 train | Broden | ImageNet val | ImageNet val + Broden | Average | Differentiable |
|---|---|---|---|---|---|---|---|
| mpnet cos similarity | cos | 0.2761 | 0.215 | 0.2823 | 0.2584 | 0.2580 | **Yes** |
| | SoftWPMI | **0.3664** | 0.3945 | **0.5257** | **0.5233** | **0.4525** | No |
| | cos cubed | 0.3579 | **0.4101** | 0.5187 | 0.5223 | 0.4523 | **Yes** |
| Top1 accuracy | cos | 8.50% | 5.70% | 15.9% | 11.4% | 10.38 % | **Yes** |
| | SoftWPMI | **46.20**% | **70.50** % | **95.00**% | **95.40**% | **76.78**% | No |
| | cos cubed | 31.00% | 49.40 % | 87.40% | 85.50% | 63.33% | **Yes** |

Table 3: Comparison of the performance between similarity functions. We look at the final layer of ResNet-50 trained on ImageNet. We use 20,000 most common English words as the concept set for mpnet cos similarity and ImageNet classes as the concept set for top1 accuracy. We can see *cos cubed* performs much better than simple *cos*, and almost as well as *SoftWPMI*.

## A.4  ABLATION: EFFECT OF CONCEPT FILTERS

In this section we study how each step in our proposed concept filtering effects the results of our method. In general our use of filters has two main aims:

- To improve the interpretability of our models
- Improve computational efficiency and complexity by reducing the number of concepts

The filters are not designed to improve performance of the models in terms of accuracy, in fact we would expect them to slightly reduce accuracy as a model with more concepts is generally larger and more powerful.

To evaluate the effect each individual filter has we trained models on both CIFAR10 and ImageNet while deactivating one filter at a time, as well as one without using any filters at all. The results are displayed in Table 4. We can see that the accuracy of our models is not at all sensitive to the choice of filters, with ImageNet accuracy remaining constant and CIFAR10 slightly increasing its accuracy with less filters. We were unable to train an ImageNet model without any filters as training with the large number of concepts required more memory than we had available in our system. Additionally the ImageNet model without similarity to other concepts filter had worse accuracy and less sparsity than other tested models, which we think may be caused by the GLM-SAGA optimizer not finding as good of a solution when the number of concepts increased too much.

| Filters | CIFAR10 Accuracy | CIFAR10 #concepts | ImageNet Accuracy | ImageNet #concepts |
|---|---|---|---|---|
| All filters | 86.26% | 142 | 71.89% | 4380 |
| No length filter | 86.55% | 146 | 71.88% | 5347 |
| No similarity to classes filter | 86.74% | 148 | 71.93% | 4671 |
| No similarity to other concepts filter | 86.42% | 151 | 70.90%* | 6515 |
| No CLIP activation filter | 86.41% | 147 | 71.92% | 4478 |
| No projection accuracy filter | 86.33% | 147 | 71.88% | 4462 |
| No filters at all | 86.56% | 177 | N/A | 9087 |

Table 4: Effect of our individual concept filters on the final accuracy and number of concepts used by our models. *We were unable to train the model to be sparse enough, results from a less sparse model.

### A.5   FURTHER DISCUSSION AND EXAMPLE OF CONCEPT FILTERING

To further show the effects of each filter, we will showcase the full filtering procedure for our CIFAR10 model with all filters. We start with a freshly generated initial concept set from GPT-3, which has a total of 177 concepts. We then take the following steps to filter down concepts:

1. **Delete concepts that are too long.** This step is important to make sure concepts are easy to visualize and simple non-convoluted concepts. For CIFAR-10 this leads to deletion of 3 concepts:
   *- white spots on the fur (in some cases)*, *feline features (e.g., whiskers, ears)* and *legs with two toes pointing forward and two toes pointing backwar.*
   174 concepts remain.

2. **Delete concepts too similar to output classes.** This step is required for the explanations to be informative, as it helps avoid trivial explanations such as: "This is image is a cat because it is a cat". This step removes the following CIFAR-10 concepts (with the class name it is too close to shown in brackets):
   - a cat (cat), a deer (deer), a horse (horse), a plane (airplane), a car (automobile), car (automobile), cars (automobile), vehicle (automobile), animal (dog), truck driver (truck)
   164 concepts remain

3. **Delete concepts too similar to other concepts.** In this step we aim to delete duplicate concepts from the concept set, i.e. two concepts that have the same semantic meaning should not both be part of the concept set. Duplicate concepts will cause unnecessary computational cost and confusing explanations. For CIFAR-10 we delete the following concepts (with the concept it is too similar to in brackets):
   - 4 wheels (four wheels), a food bowl (a bowl), a furry, four-legged animal (a large, four-legged mammal), a gasoline station (a gas station), a large boxy body (a large body), a large, muscular body (a large body), a street (a road), a strong engine (an engine), large, bulging eyes (large eyes), the ocean (the sea)
   154 concepts remain

4. **Remove concepts not present in training data.** In this step we remove all concepts that CLIP thinks are not really present in training data. The purpose is to avoid learning neurons that don't correctly represent their target concepts, as concepts missing from the dataset are unlikely to be learned correctly in the Concept Bottleneck Layer. For CIFAR-10 this step deletes the following concepts:
   - a bit, a mechanic, legs, long legs, passengers, several seats inside, windows all around
   147 concepts remain

5. **Remove concepts we can't project accurately.** The purpose of this step is similar to the previous step, we want to remove all concepts that are not faithfully represented by the CBL. To do this we evaluate the similarity score between the target concept and the activations of our new neuron using CLIP-Dissect Oikarinen & Weng (2022) on the validation data, and delete concepts with similarity less than 0.45 which was determined to be a good indicator of faithful concept representation. For CIFAR-10 we delete the following concepts:
   - a bed, a coffee mug, a crew, a house, a wheel
   We are left with our 142 final concepts.

In total our method has 5 different cutoff parameters that can be tuned. Since the main purpose of these filters is to make the network more interpretable, their values were mainly chosen through trial and error to find ones that produce the most useful and explainable models. Most of these cutoffs are independent of the dataset chosen, and the only one we changed from one dataset to another was the value of cutoff for images being present in the dataset (filter 4). We used a cutoff of 0.25 for CIFAR-10 and CIFAR-100, 0.26 for CUB-200 and 0.28 for Places and ImageNet. We found we had to change this cutoff as CLIP doesn't activate very highly on low resolution images of CIFAR, and in general we get higher top5 activations on larger datasets. Since the other 4 cutoffs are fixed, using our method for a new dataset won't require a large hyperparameter search, and as seen in Table 4 the overall performance is not very sensitive to specific filters.

To visualize the effect of our filters, in Figure 7 we visualize the final layer weights for CIFAR-10 class "automobile" trained with and without filters. We can see the model with filters has found a

reasonable decision rule, with largest weights corresponding to *four wheels* and *a steering wheel*. On the other hand, the weights for the model without filters are quite problematic. First, the 3 largest weights are *cars*, *car* and *a car*. Not only do they have the same meaning as the final class itself (filter 2), thus lacking any explanatory power, they are duplicates of each other (filter 3), making the explanation unnecessarily compilicated. Finally we see the unrelated concept *bed* which is not present in CIFAR-10 having a small positive weight, which should be removed by either filter 4 or 5.

Figure 7: *Comparison of the final layer weights between CIFAR-10 model trained with and without filters.*

## A.6   ABLATION: EFFECT OF INITIAL CONCEPT SET

In this section we evaluate the effect our initial concept set generator has on our results. In our paper we proposed a new method of generating descriptions using GPT-3 Brown et al. (2020), and in this section we compare those results to the case where we generate the initial concept set using ConceptNet Speer & Havasi (2013) as proposed by Yuksekgonul et al. (2022). In general ConceptNet only works well when prompted with relatively common single word phrases, but many of class names for CUB200, Places365 and ImageNet consist of multiple words, such as the ImageNet class *great white shark*. To overcome this issue we split multi-word class names into single word components and add concepts related to each word to the concept set, in this case *great*, *white* and *shark*. After creating the initial concept set we use the same filtering procedure described in section 3.1B for both models. Table 5 displays the accuracy comparison between models trained on different initial concept setst. In general using GPT3 seems to provide a small (0.1-1.5%) accuracy boost on all datasets, with the exception of CUB200 where the ConceptNet model fails completely. This is because ConceptNet is unable to generate concepts for the highly specific class names in CUB200, such as *Groove billed Ani*, while GPT-3 is not troubled by this.

| Model\Dataset | CIFAR10 | CIFAR100 | CUB200 | Places365 | ImageNet |
|---|---|---|---|---|---|
| LF-CBM (GPT-3) (original) | **86.40%** | **65.13%** | **74.31%** | **43.68%** | **71.95%** |
| LF-CBM (ConceptNet) | 86.30% | 63.62% | 2.19% | 42.49% | 71.52% |

Table 5: Accuracy comparison between using different methods for generating initial concept set.

## A.7 ABLATION: LF-CBM WITHOUT SPARSITY CONSTRAINT

To estimate the effect sparsity has on our models, we show the accuracy of LF-CBM models trained without sparsity constraints (LF-CBM (dense)) in Table 6. We can see that not using a sparsity constraint greatly improves accuracy (except for CUB200 which starts to overfit), but it comes at a very large cost for interpretability as can be seen in Figure 8, where almost none of the decision is explained by the 10 most highly contributing concepts.

| Model\Dataset | CIFAR10 | CIFAR100 | CUB200 | Places365 | ImageNet |
|---|---|---|---|---|---|
| LF-CBM (sparse [original]) | 86.40% | 65.13% | 74.31% | 43.68% | 71.95% |
| LF-CBM (dense) | 87.50% | 67.93% | 74.25% | 48.25% | 74.09% |

Table 6: Results of training our CBM without sparisity constraints on the final layer.

Figure 8: Dense CBM explanation for the same image as the fourth image in Figure 17. We can see the dense model is practically uninterpretable.

## A.8 CONSISTENCY OF CONCEPT SET GENERATION

Since the initial concept set is generated using GPT-3, it is a random process and in this section we explore how much this noise effects our final results. When regenerating initial concept set and running our concept filtering pipeline for our ImageNet model, we got 4380 concepts, compared to the original 4523 concepts. In terms of accuracy, the model learned with regenerated concept set reached an accuracy of 71.89%, compared to the average of 71.95% with the original concept set. This indicates that the difference is very small in terms of accuracy. However, the exact concepts used and model weights can be pretty different as shown in Figure 9, despite all being relevent (e.g. there are many concepts that appear on both concept sets, e.g. Image 49962 has important concepts of "wipes", "toiletries" on both, though they have different corresponding contributions). Figure 9 shows the explanations for decisions on two random images for the original model and model trained with regenerated concept set.

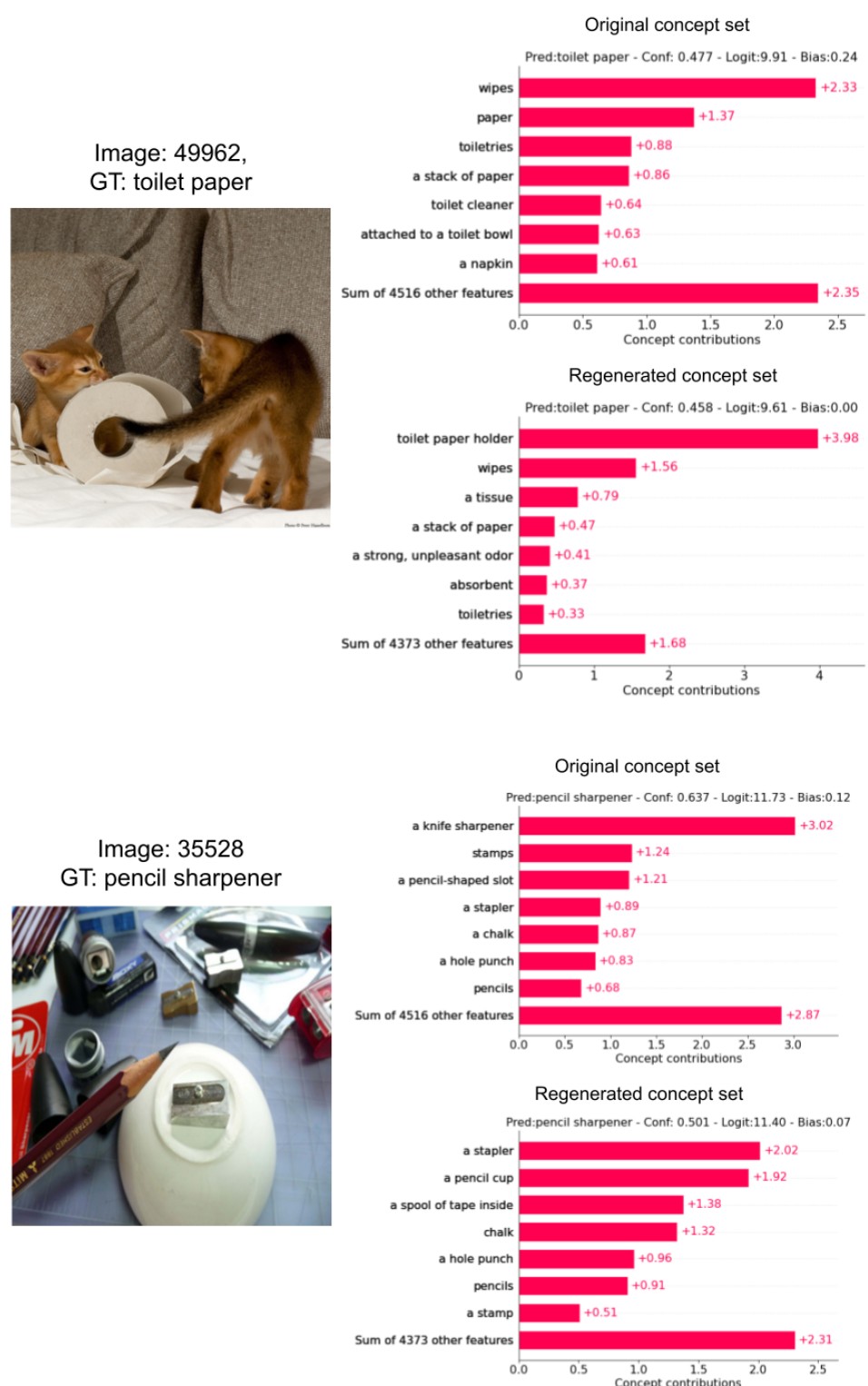

Figure 9: Difference in concepts used for a decision and their weights for models trained with original vs regenerated concept set, shown on two random images.

### A.9 MORE DETAILS ON MODEL EDITING PROCEDURE

Here we provide an improved explanation of Sec 5.2 on how we can edit the final layer weights of our proposed Label-Free CBM to correct incorrect predictions. Our procedure includes 3 steps:

**Step 1: Find an input image $x_i$ where the model makes a Type 4 Error (incorrect final layer weight)**:

This is done by visualizing incorrect model predictions and their explanations similar to Figure 4. To identify if it is a Type 4 error, first we check that the ground truth label of the input image is correct and unambiguous (therefore not type 1 error described in sec 5.1). Next, we check that the highly activating concepts look correct (therefore not type 2 or type 3 error described in sec 5.1).

**Step 2: Identify a concept to edit**:

We start by listing the highest activating concepts for the chosen input, i.e. the highest elements of $f_c(x_i)$ (after normalization). This allows us to identify concepts that are the most important for this specific image while not too important for other images. From these concepts we use our domain knowledge (and/or internet search) to understand which concepts are relevant to making this decision.

For example, the image in the right panel of Figure 6 is originally predicted as a "Shopping basket", while the ground truth is a "Hamper". The 5 most highly activating concepts for this image are: "a basket", "made of rope or string", "a rope", "a laundry basket" and "a fishing net". After a short investigation we find that although the classes "Hamper" and "Shopping basket" are very similar, hampers are more often constructed in a woven like manner. Therefore, we identify the concept of "made of rope or string" as relevant for capturing this difference, and choose it for editing in the next step.

**Step 3: Change weights by sufficient magnitude**:

Once we have selected the concept to be corrected, we will compute how much the magnitude of the associated final layer weight should be changed (denoted as $\Delta w$, $\Delta w \in \mathbb{R}$) and then edit the weights in the following way:

$$W_{F[gt,concept]} \leftarrow W_{F[gt,concept]} + \Delta w, \quad W_{F[pred,concept]} \leftarrow W_{F[pred,concept]} - \Delta w \quad (3)$$

Note that $W_{F[i,j]}$ denotes the $(i,j)$ element in the matrix $W_F$. The goal of editing is to correct the inaccurate prediction on this specific instance while minimizing effect on other predictions. To calculate $\Delta w$ needed to flip prediction, we first calculate the difference in logits $\Delta a$ (before softmax) before the edit, $\Delta a = W_{F[pred,:]} f_c(x_i) - W_{F[gt,:]} f_c(x_i)$.

Since we would like the prediction to be corrected to gt class, our goal is to design $\Delta w$ such that

$$(W_{F[gt,:]} + \Delta w \cdot e) f_c(x_i) - (W_{F[pred,:]} - \Delta w \cdot e) f_c(x_i) > 0 \quad (4)$$

where $e$ is a one-hot row vector with the entry $e_{[concept]} = 1$. Let $b = (W_{F[gt,:]} + \Delta w \cdot e) f_c(x_i) - (W_{F[pred,:]} - \Delta w \cdot e) f_c(x_i)$, where $b$ is a nonnegative constant deciding how large of a margin we want the correct prediction to have. Since $\Delta a = W_{F[pred,:]} f_c(x_i) - W_{F[gt,:]} f_c(x_i)$, we have:

$$2\Delta w \cdot f_c(x_i)_{[concept]} - \Delta a = b \Rightarrow \Delta w = (\Delta a + b)/(2 f_c(x_i)_{[concept]}) \quad (5)$$

Typically we use values for $b$ between 0.2 and 2 to calculate required $\Delta w$.

It is worth noting that when the model weights are edited, it might affect other image's predictions too due to the change of weight parameters. Thus, with the goal to correct the wrong predictions while not affecting other already correct predictions, we suggest each edit should be checked with a validation dataset before applying on a target model. As described in the end of Sec 5 in the manuscript, with the above proposed model weight editing, we are able to improve the overall model accuracy from 71.98% to 72.02% by performing the model editing for only 5 different images, which is a non-negligible improvement. Since the edits only affect weights for 10/1000 classes, this corresponds to a 4% accuracy boost on the affected classes.

## A.10 ADDITIONAL FIGURES

Figures 10, 11 provide examples of the full prompts we used for GPT-3, as well as GPT outputs. For all experiments we used the text-davinci-002 model available through OpenAI API.

Figure 12 shows the additional model edits performed in our ImageNet CBM experiment, and Figure 13 showcases Type 1 and Type 2 errors made by our ImageNet CBM.

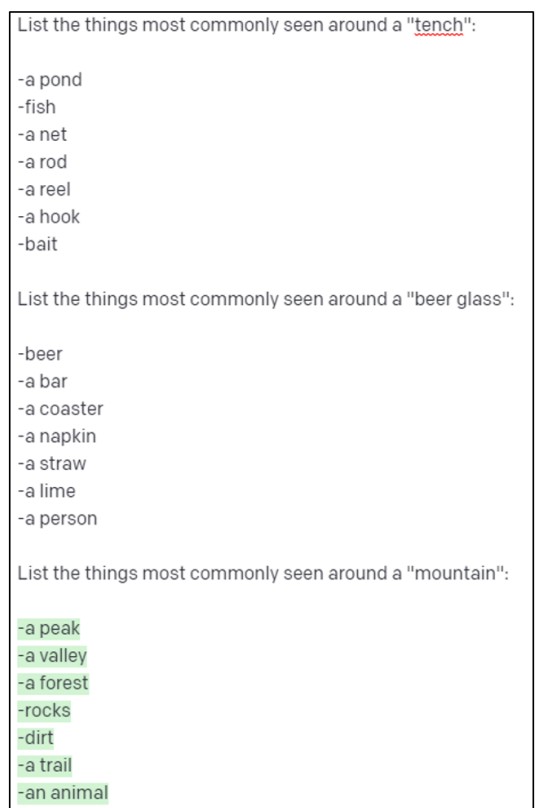
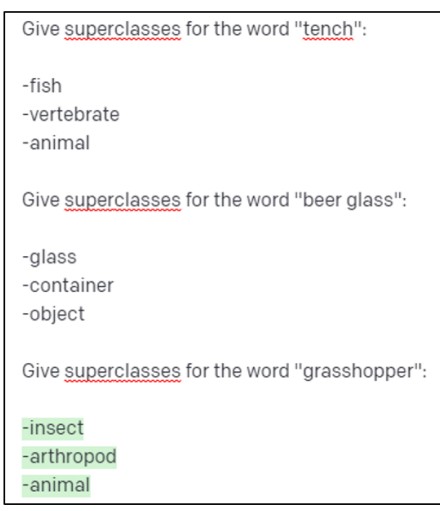

Figure 10: Full prompts used for GPT-3 concept set creation. Text in green generated by GPT, rest is our prompt.

List the most important features for recognizing something as a "goldfish":

-bright orange color
-a small, round body
-a long, flowing tail
-a small mouth
-orange fins

List the most important features for recognizing something as a "beerglass":

-a tall, cylindrical shape
-clear or translucent color
-opening at the top
-a sturdy base
-a handle

List the most important features for recognizing something as a "scuba diver":

-a person wearing a scuba mask and breathing apparatus
-a wet suit
-flippers
-a diving tank

Figure 11: Full prompts used for GPT-3 concept set creation. Text in green generated by GPT, rest is our prompt.

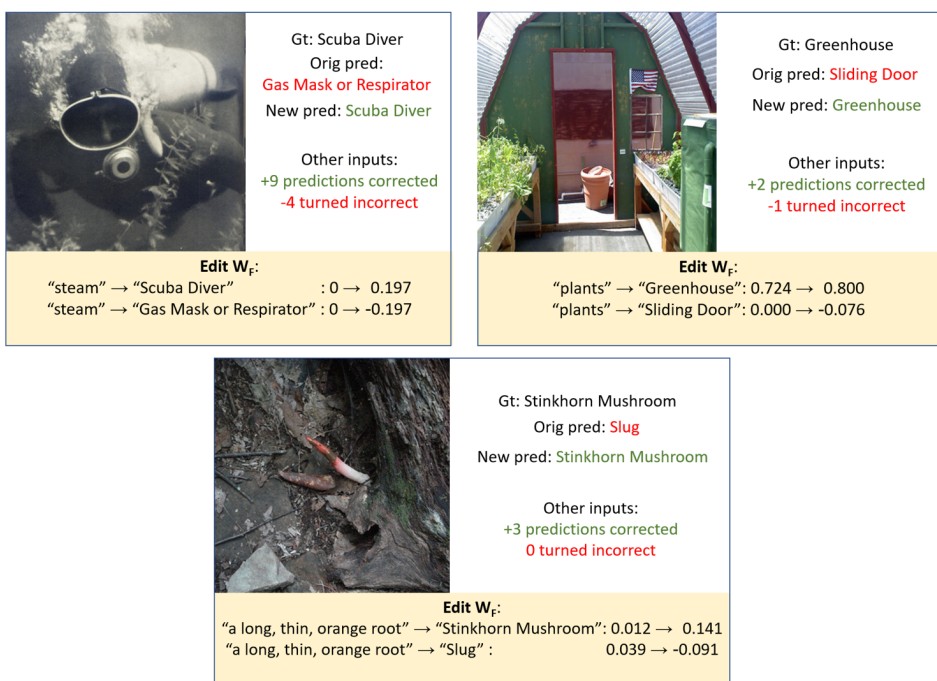

Figure 12: The other 3 model edits we performed in our experiment

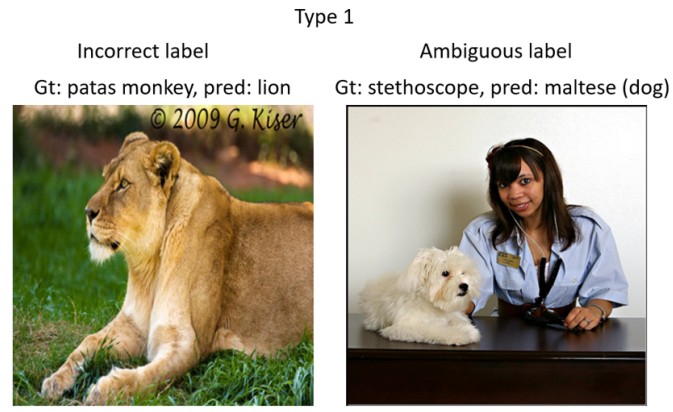

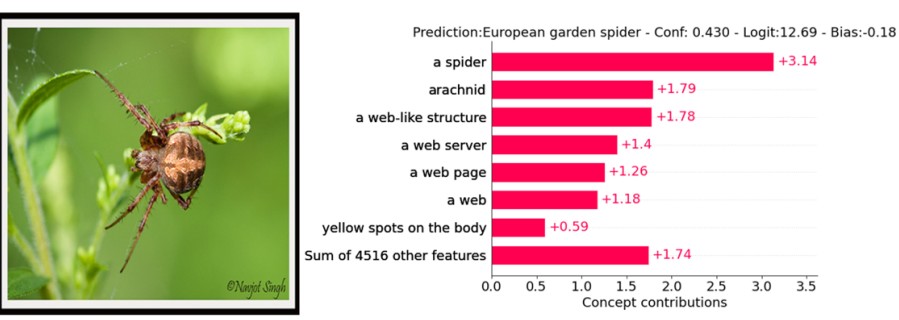

Figure 13: Examples of Type 1 and type 2 model errors.

## A.11 ADDITIONAL WEIGHT VISUALIZATIONS FOR RANDOM CLASSES

In figures 14 and 15 we have visualized the final layer weights leading to 4 randomly chosen output classes for both Places365 and ImageNet. These were created using the same procedure as Figure 3, with minor differences such as focus on single class at a time and no manual editing of concept colors according to semantic similarity. We can see that the concepts that have large weights are indeed relevant to the class, and overall the weights look reasonable.

Figure 14: Weight visualizations for 4 randomly chosen output classes for our CBM trained on Places365.

Figure 15: Weight visualizations for 4 randomly chosen output classes for our CBM trained on ImageNet, as well as an example image of each class to clarify their meaning.

### A.12 Additional explanations for random images

In figures 16, 17, 18 we display additional samples of decision explanations for 4 randomly chosen input images for each of Places365, ImageNet and CUB-200 respectively. Overall, it can be seen that the concepts and explanations are of good quality, even with the CUB-200 dataset which may require very specific knowledge and concepts to create CBMs. This result demonstrates the effectiveness of our LF-CBM, which does not use the expert-knowledge concept sets unlike existing CBMs.

# Places365

Image:19587, Ground truth: Jacuzzi, indoor

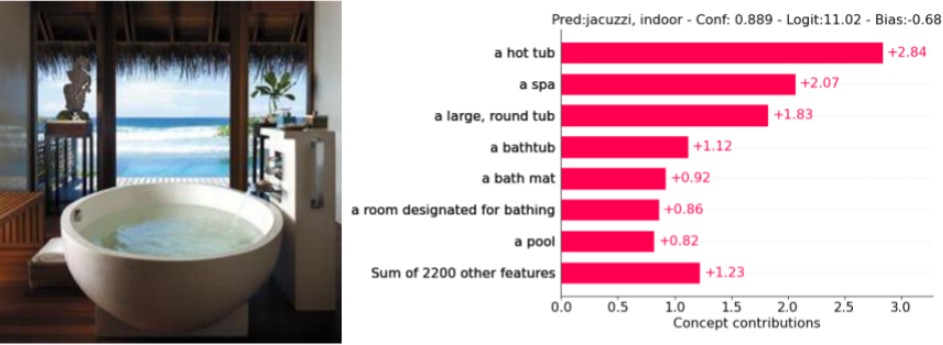

Image:5776, Ground truth: boardwalk

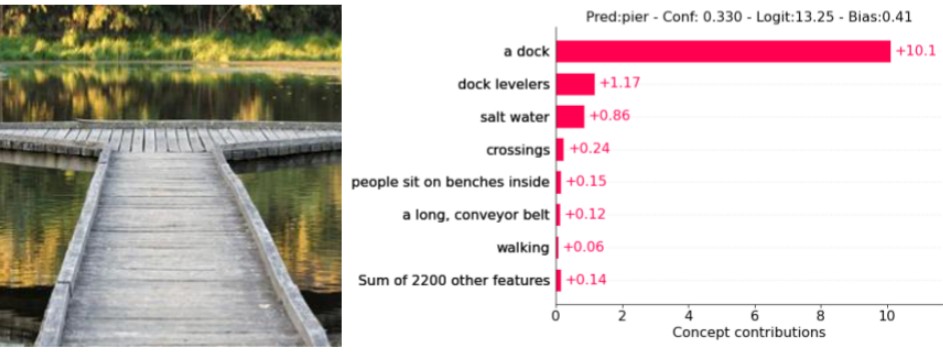

Image: 17442, Ground truth: heliport

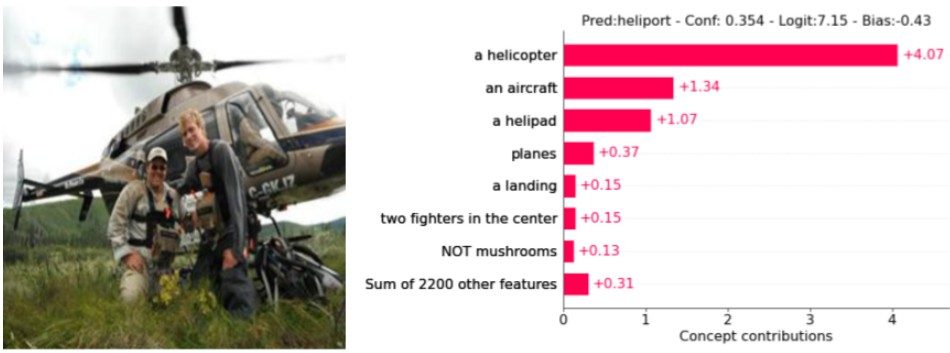

Image: 33310, Ground truth: Topiary Garden

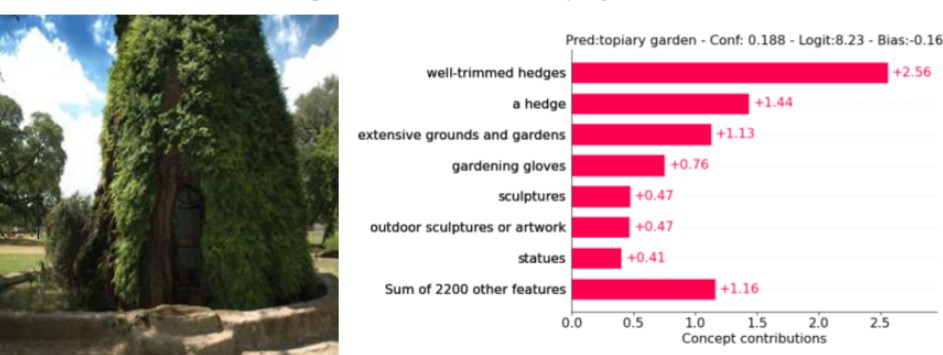

Figure 16: Explanations for 4 randomly chosen input images for our CBM trained on Places365.

# ImageNet

Image:17237, Ground truth: hippopotamus

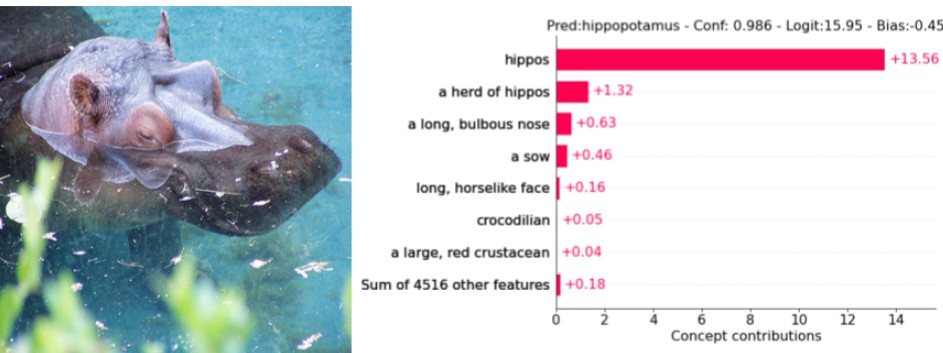

Image:33159, Ground truth: monastery

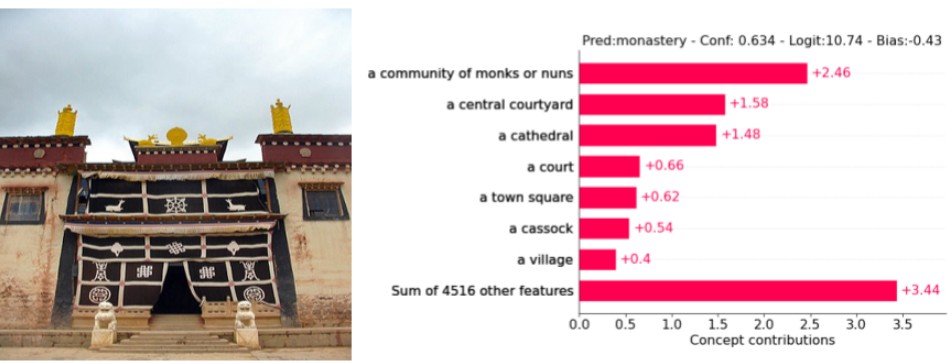

Image: 23074, Ground truth: breastplate

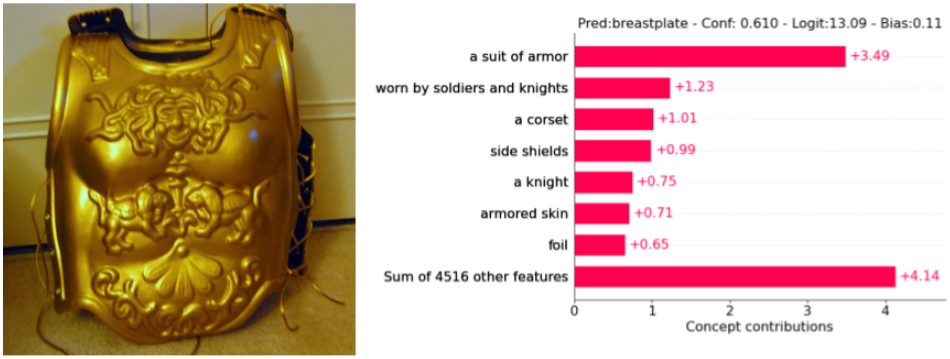

Image: 13260, Ground truth: Toy Poodle

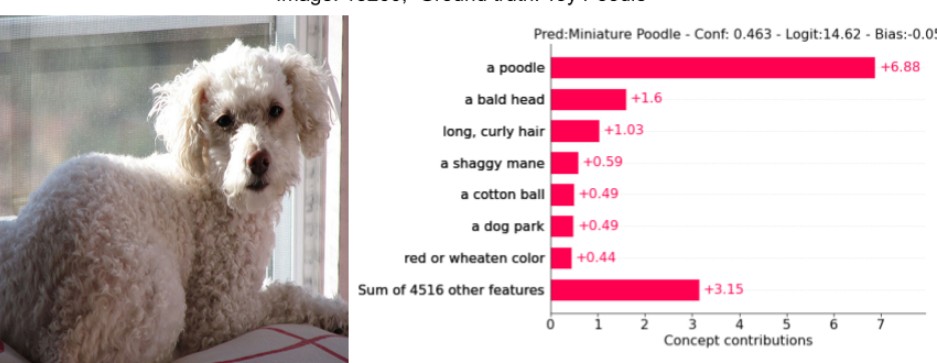

Figure 17: Explanations for 4 randomly chosen input images for our CBM trained on ImageNet.

# CUB200

Image:1169,  Ground truth: Yellow bellied Flycatcher

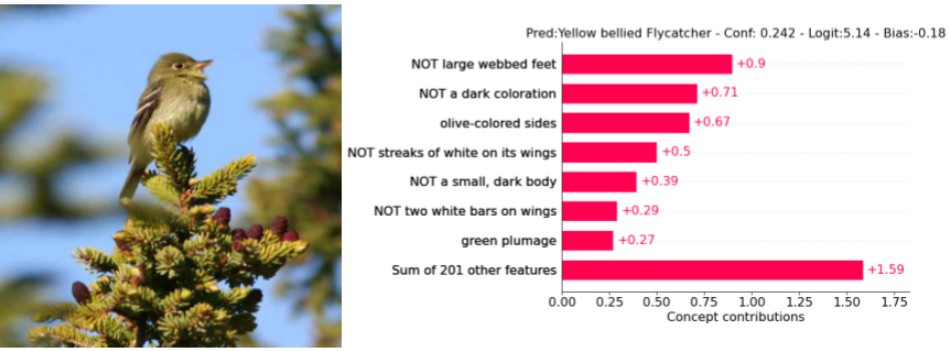

Image:4121,  Ground truth: Black Tern

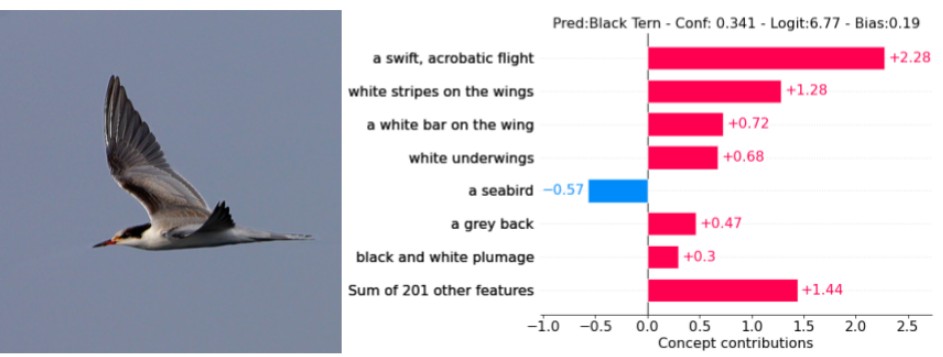

Image: 5159,  Ground truth: Tennessee Warbler

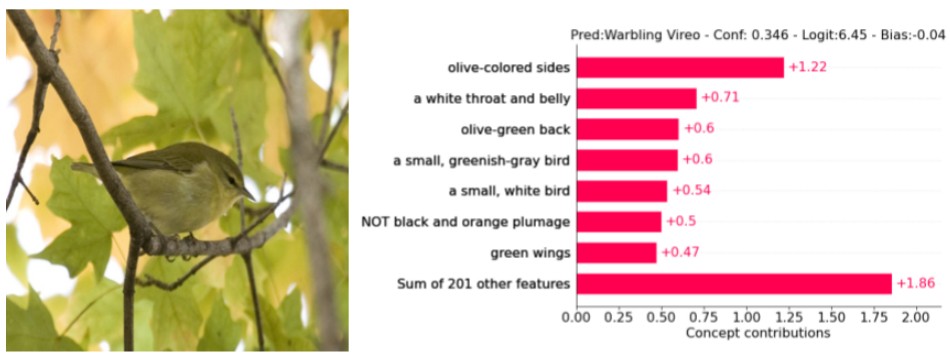

Image: 4823,  Ground truth: Kentucky Warbler

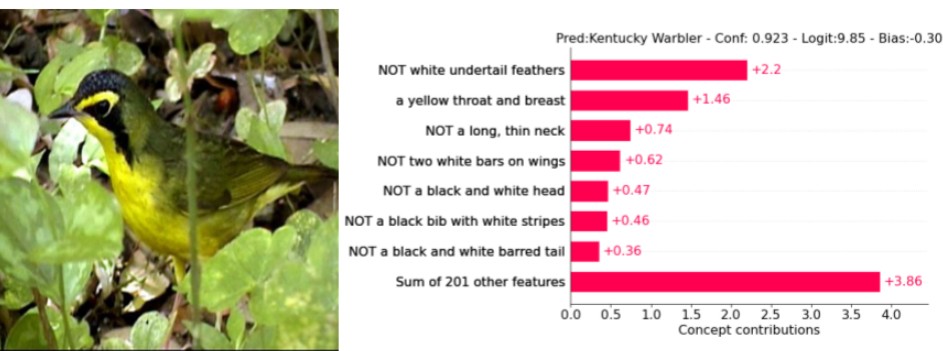

Figure 18: Explanations for 4 randomly chosen input images for our CBM trained on CUB-200.

