# OpenReview forum: "Label-free Concept Bottleneck Models"
_ICLR.cc/2023/Conference — ICLR 2023 poster_

### Official Review · Reviewer_8fC7 · 2022-10-24

**Confidence:** 3
**Correctness:** 2
**Technical Novelty And Significance:** 3
**Empirical Novelty And Significance:** 2
**Recommendation:** 6

**Clarity, Quality, Novelty And Reproducibility:**

**Clarity**

The paper is well-written, clear and easy to follow.

**Quality**

The scope of validity of the proposed solution is unclear to me. Additionally, an ablation study is required in order to gain insights on the impact of the different training stages on the final predictive performance. Finally the quantitative experiments on accuracy should follow the original work of P-CBM.

The qualitative analysis with the visualisations in Section 4.4 and the use case in Section 5 are original and informative.

**Novelty**

The problem of automating the process of concept definition is to my knowledge new and relevant. Also, the idea of reusing existing technology (i.e. GPT-3) for tackling such problem is novel and original.

**Reproducibility**

Code is not available. However, the authors are willing to release it prior to publication.

**Strength And Weaknesses:**

**Strenghts**
- The paper is clear and easy to follow.
- The idea is simple, intuitive and original. I appreciated the idea of reusing existing technology and to apply it to concept bottleneck models.
- The visualisations in the experiments are compelling and informative.

**Weaknesses**
- The proposed strategy includes several heuristics and an ablation study is missing.
- It’s unclear what is the scope of application of the proposed strategy, as it seems to be limited to traditional vision benchmarks. For instance, what happens if classes are imbalanced or if data come from specific domains?
- The experimental evaluation (Section 4.2) is non-standard and the authors should consider the same experiments performed by post-hoc networks. Additionally, the authors should report the set of whole results in the original paper.
- From a computational point of view, the proposed strategy is quite complex.
- Code is not available. However, the authors have stated that they will release it prior to publication.

**Detailed comments and constructive feedbacks**
1. Several heuristics are used throughout the different stages of training. For instance, in stage 1 several rules are used to filter out the generated concepts.  How are these rules and hyperparameters affecting the final performance? Are these configurations data/task dependent?
2. At which stage is the main backbone trained?
3. The experimental analysis focuses on 5 traditional vision datasets, namely CIFAR10, CIFAR100, CUB200, Places365 and ImageNet. I can imagine that the concept generated by GPT-3 can transfer to these benchmarks and therefore help in the automation process. However, what does it happen when the downstream tasks include rare classes (e.g. iNaturalists) or it requires concepts which requires domain specific knowledge (e.g. medical data)?
4. Why aren’t the experiments on accuracy following the ones in the paper of post-hoc concept bottleneck models (P-CBM)? The author should start from the results in Table 1 of the paper of P-CBM, rather than running different experiments.
5. In Section 5.2, only 5 interventions are found to be useful. Are there any examples of interventions in Section 5 which hamper the predictive performance? And how hard is to find such beneficial interventions?
6. What are the limitations of the proposed methodology?

**Additional missing references**

[2] Concept Whitening for Interpretable Image Recognition. Nature Machine Intelligence 2020
[3] Concept Embedding Models. NeurIPS 2022

**Summary Of The Paper:**

The paper introduces a methodology to train concept bottleneck models, which automates the process of concept definition and it is empirically shown to outperform existing baselines making use of user-defined concepts. The methodology consists of four stages:
In stage 1, concepts are defined by generating lists of words conditioned on information related to each class in the downstream task using GPT-3. Additionally, a set of heuristics is applied to filter out generated concepts and to produce a curated concept list which is used in the subsequent stages. In stage 2, CLIP embeddings for the image and the concepts are used to produce a similarity matrix. In stage 3, the concept bottleneck layer is trained to maximise a similarity score between the weights of the layer and the similarity matrix. Finally, In stage 4, a linear layer is learnt to solve the downstream task.

Experiments are conducted on 5 common vision benchmarks, including CIFAR10, CIFAR100, CUB200, Places365 and ImageNet, by comparing the proposed strategy against post-hoc concept bottleneck models [1] and a black-bock baseline. Additionally, the work proposes a new strategy intervening on the weights of the final prediction layer as demonstrated with a use case on ImageNet.

[1] Post-hoc Concept Bottleneck Models. ICLR 2022 PAIR2Struct Workshop

**Summary Of The Review:**

I recommend for an initial score of 5, as the weaknesses are currently outweighing the strengths.

----- POST REBUTTAL ------

I went through the rebuttal. The authors have addressed many of my concerns, (i) by including an ablation analysis on concept filtering, (ii) discussing the limitations of the proposed approach and (iii) providing substantial clarifications to all my questions.
The quality of the paper has consequently increased thanks to the ablation analysis. The scope of the proposed solution has been made clearer thanks to the discussion on the limitations.
Overall, I'm quite positive about the work. The idea is novel and original, pushing forward research on explainability for concept bottleneck models. The only drawback is related to the reproducibility. Certainly, building solutions around proprietary software (like GPT-3) is not a good practice towards open science. However, the authors plan to release their code, all generated concepts and also make the necessary steps to ensure that all experiments are reproducible.

Based on these considerations, I increase my score.

---

> ### Author Response · Authors · 2022-11-19
> **Author response 1/2**
>
> R4
> Thank you for the review and constructive feedback! Please see our response below for your questions and comments.
>
> **#1 Ablation study on Concept filtering**
>
> Thank you for pointing this out! Following your suggestions, we have conducted a large ablation study on the effect of our concept filters, and discussed them and the reasoning behind them further in Appendix B.1. We can see that the performance of our networks in terms of accuracy is not sensitive to the filters used at all, but they are important in improving the interpretability and computational efficiency of our models. In Figure 11 we can see that without using filters, our predictions are still somewhat reasonable but may suffer from issues such as explaining the prediction by the class itself, i.e. *this image is and automobile because it is a car*, and having duplicate concepts such as *car* and a *a car* which makes explanations confusing. Please see our Appendix B.1 for full details and results, including a walkthrough of how the concepts are used and which concepts are removed by each filter for our CIFAR-10 model. Regarding number of cutoffs and thresholds, we agree there are many associated with the filtering (5 total), however 4 of those are fixed for all datasets/tasks and as such do not provide a challenge for automation/future application. We hope this addresses your concerns and please let us know if you have any remaining questions.
>
>
> **#2 Scope of application of the study**
>
> Thank you the comment, we have added a discussion about the scope of application for method in a newly generated Limitations section in Appendix A.1, saying:
> “Our model is aimed to extend Concept bottleneck models to large datasets where labeled concept data is not available. However it works best on domains where CLIP performs well and as such may not perform as well on small datasets that require specific domain knowledge and labeled concept data is available, such as medical datasets. For such datasets we believe it is still better to use a method that can effectively leverage the labels available such as original Concept Bottleneck Models or P-CBM.”
>
> However we did show our model performs well on CUB200 which requires specific knowledge about bird species, and it is possible that it would perform well on other specific datasets like medical images, but we did not have time to run an experiment on that during this rebuttal. We do not see why imbalanced classes would cause issues to our method beyond the issues they cause most in deep learning approaches.
>
> **#3 Non-standard evaluation**
>
> We do not believe that there exist a standard set of evaluations for this kind of paper. For example the P-CBM paper (Yuksekgonul et al, arxiv 2022) evaluates their models on CIFAR-10, CIFAR-100, CUB, HAM10000 and ISIC. In the meantime, the original Concept Bottleneck Models paper (Koh et al, ICML 2020) evaluates their results on CUB and OAI, and the Interpretable Basis Decomposition paper (Zhou et al, ECCV 2018) evaluates their methods on Places365. It can be seen that the datasets that they used to evaluate their results have almost no overlap, except the CUB dataset.
> On the other hand, we evaluated on 5 datasets that include at least one dataset in each paper, with 3 datasets that are reported in P-CBM. Note that we already reported results on 3/5 datasets that P-CBM report, while P-CBM does not have code available yet. Also note that the remaining two dataset used in P-CBM are medical datasets with labeled concepts available, which is does not require a *Label-Free* CBM, and our method is aimed at datasets where the labeled concepts are not available (e.g. CIFAR, ImageNet). Thus, our results mostly focused on this type of dataset, showcasing the flexibility of Label-Free CBMs.
> Furthermore, as discussed in previous response and section A.1 of the updated manuscript, the goal of our method is not to replace existing models in these settings, but instead to enable using CBMs on large datasets without labeled concept data available, which is why we have focused more on large scale experiments with Places365 and ImageNet.

---

> ### Author Response · Authors · 2022-11-19
> **Author response 2/2**
>
> **#4 Computational complexity**
>
> Although our method includes many steps, most of them are actually not computationally expensive. In fact, most of the computational cost is running the training dataset through the backbone model, which can be done just once. The training part is pretty light, as the only trainable parameters are the concept bottleneck layer W_c and final layer W_F. As already reported in the sec 4.1 in the original manuscript, training on a single Nvidia Tesla P100 GPU, the full training run only takes a few minutes (cifar model) to 20 hours (imagenet, place365), which is pretty efficient. In addition, we have evaluated the importance of different parts of our pipeline in our rebuttal experiments, as described in Appendix sections B.1, B.2 and B.6. Finally, our pipeline performs cleanly together and can be executed without a large number of lines of code.
>
> **#5 When is the main backbone trained**
>
> As in Post-hoc CBM, the backbone f(x) is a pre-trained model, and training it is not a part of our pipeline.
>
> **#6 Interventions in section 5.2**
>
> Yes, some interventions turn out to be harmful to the overall performance as in general the weights are already trained to be close to optimal and most changes reduce performance. However, our method can quickly find interventions that help performance and finding these 5 did not take very long at all. Since some interventions are harmful, we highlight the need use a validation dataset when making these interventions for real life use cases, i.e. first, you find an opportunity for intervention, then evaluate its effect on a validation dataset, and if the effect is good deploy it on the production model.
>
> **#7 Limitations and additional references.**
>
> We have added a section describing the limitations of our model in Appendix A.1. Thank you for the suggestions we have included your references in section 2.
>
> **#8 New experiments and changes made in this rebuttal period**
>
> We have made many clarifications to the manuscript and performed many new experiments, please see our General Response: Overview of new results and changes in a separate post for a summary of all these changes.
>
> **#9 Summary**
>
> - #1 Performed a large ablation study on the effects of concept set filtering, showing our model is not very sensitive to changes in these filters but also showcased the interpretability benefits of using them
> - #2 Discussed the scope of application of our study and added new section discussing limitations to the manuscript
> - #3 Discussed the differing aims between our method and P-CBM and why our evaluation is not the same
> - #4 Addressed concerns regarding computation complexity
> - #5 Clarified that backbone networks are pre-trained
> - #6 Discussed further our procedure for editing final layer weights
> - #7 Discussed the limitations of our method and added sugested references
> - #8 Discussed our other new experiments and changes made to the manuscript during the rebuttal period
>
> We believe that we have addressed all your concerns. Please let us know if you still have any reservations and we would be happy to address them!

---

> ### Author Response · Authors · 2022-11-29
> **Thank you**
>
> Thank you for the updated review, we are happy you find the new submission stronger!

---

### Official Review · Reviewer_mHS9 · 2022-10-25

**Confidence:** 4
**Correctness:** 2
**Technical Novelty And Significance:** 3
**Empirical Novelty And Significance:** 3
**Recommendation:** 6

**Clarity, Quality, Novelty And Reproducibility:**

The paper is mostly clearly written, though with some unsupported claims.  Prompting an LM for concepts followed by filtering is not entirely novel (Yejin Choi’s group has quite a bit of work in the area ), but the combination of the methods is novel.
Reproducing would probably not be possible since prompting a LLM isn’t entirely deterministic so its unclear I’d get the same concept set as the authors ( and they aren’t releasing code or the concepts they generate ) , but its likely I’d get approximately similar results ( though not guaranteed ).


**Strength And Weaknesses:**

**Strengths:**
The use of CLIP to generate task specific concepts sets seems like a useful method and the use of CLIP-Dissect (CD) and the GLM-SAGA solver also seem novel and well motivated.

**Weaknesses**
The filtering methods felt like heuristics that should have included ablation studies to justify certain decisions ( concepts not being near classes for instances, and all the  thresholds that are included in section 3.1 B ).

The paper makes a few unsupported claims such as
1) in general we found GPT-3 concepts to be of higher quality (than ConceptNet), but you never actually do a qualitative comparison or study in the paper to back that ( and a concept quality analysis is one of the major things missing from the paper).   Using a human curated ontology would seem to have some advantages over task specific concept sets ( namely using the same concepts over different data and tasks) though the paper shows accuracy wise they get better result, but this would have been made stronger had they included an experiment showing the results they would have obtained by using Concept Net to generate their concept set ( ie, how much is it the concepts vs the architecture which gives this paper a performance bump.
2) criticizing the use of CLIP in Posthoc CBMs when its a substantial part of the architecture here  ( I honestly could be missing something, but it doesn’t seem like Post CBMs tie them to CLIP anymore than the Label Free model is? )
3) In Table 2 you show Label-Free CBM  outperforms Post-hoc CBM in 3 of 5 and 2 of 5 tasks for the CLIP variant . You can show values for other datasets if you have them, but without that you can’t say your work clearly outperforms PCBMs “on all datasets”.
4) Section 5 starts, “this is the first example of manually editing a large neural network with no obvious flaws”.  This is a vague assertation. What methods of manually editing are you referring to that have obvious flaws? PCBMs propose model editing in their paper so if you are commenting on their proposal or another papers proposal you should specify why they have obvious flaws.

The paper under explains important elements of the paper, in particular with regards to having human validation of the filtered concepts.
Ablations and some under explained areas include:
1) How important are each of the filtering techniques really?  An ablation on these, in particular points 1 + 2, would be useful.  The number of cutoffs and thresholds makes this feel not very “automated”.
2) What would have happened without the sparseness  constraint on the last layer in terms of accuracy?
3)  How are the decision rules  in Fig 3 learned? Is this showing the average Wf layer for training examples of each class and then showing relative weight.  This vizs are nice, but you should explain this better ( maybe swap 4.4 and 4.3 in order and then add additional explanation )
4) Its possible I’m missing something, but the algorithm in section 5.2 is quite vague and not terribly convincing because of it.  It seems like its really flipping model weights and hoping more examples are corrected than made incorrect by the new model and is not a systemic, principled way to do things.

**Nitpicks:**
1) Why do you spend time introducing IBDs as interpretable CBMs only to not compare against them because you say they have non-interpretable components?
2) I personally think its ok to have a residual layer to improve CBM performance ( PCBM-h ) and it’d be better to include those results with that caveat of the model not being fully interpretable
3) For Fig 2: you should show
- there are M concepts in concept set ( which is implied by numbers in Concept Matrix ) AND
- f_c(x) = Wc * f(x) AND
- the output of step 4 are classes and not concepts ( so add that each x has an associated y in R^p for instance )
   so the architecture its not misunderstood to be completely unsupervised.
   you then need to write that W_f is M by p

**Summary Of The Paper:**

Citing two limitations of Concept bottleneck model (CBM),  (1) the time consuming and labor intensive need to collect labeled data for each of the predefined concepts and (2) their accuracy s often significantly lower than that of a standard neural network, the authors propose Label-free CBM; a framework to transform any neural network into an interpretable CBM without labeled concept data, while retaining nearly the same accuracy as the original model.  They prompt CLIP based on templates filled in with task class names to generate an initial concept set (ie, label free) and then filter that down based on many heuristics.  They then learn a concept matrix ( the dotproduct of task image encodings with each concept text embedding ) and subsequent concept bottleneck layer of concept weights for a given image and then finally learn a sparse final layer using elastic net penalty and solved via GLM-SAGA ( from Wong et al 21 ).  They compare there results with CBMs and PostHoc CBMs (a similar-ish prior method).  They additionally show how to explain individual instances via concepts ( like CBMS ), how to generate global explainability rules from the concept weights and final layer weights and discuss model editing, namely the final sparse layer weights.

**Summary Of The Review:**

The use of CLIP to generate task specific concepts sets seems like a useful method, though the filtering methods felt like heuristics that should have included ablation studies to justify certain decisions ( concepts not being near classes for instances, and all the  thresholds that are included in section 3.1 B.   The use of CLIP-Dissect (CD) and the GLM-SAGA solver also seem novel and well motivated though I’m not sure of what the exact difference between CD and what they propose in 3.2 is.   The paper makes many unsubstantiated claims, under explains important elements of the paper, in particular with regards to having human validation of the filtered concepts and I don’t find section 5.2  terribly convincing.  The paper is a useful contribution, but could be made much stronger by addressing some of the issues and performing certain ablations to justify decisions/claims made..

---

> ### Author Response · Authors · 2022-11-19
> **Author response 1/2**
>
> Thank you for the review and useful feedback! Please see our response below for your questions and comments.
>
> **#1 Ablation study on Concept filtering**
>
> Thank you for pointing this out! Following your suggestions, we have conducted a large ablation study on the effect of our concept filters, and discussed them and the reasoning behind them further in Appendix B.1. We can see that the performance of our networks in terms of accuracy is not sensitive to the filters used at all, but they are important in improving the interpretability and computational efficiency of our models. In Figure 11 we can see that without using filters, our predictions are still somewhat reasonable but may suffer from issues such as explaining the prediction by the class itself, i.e. *this image is and automobile because it is a car*, and having duplicate concepts such as *car* and a *a car* which makes explanations confusing. Please see our Appendix B.1 for full details and results, including a walkthrough of how the concepts are used and which concepts are removed by each filter for our CIFAR-10 model. Regarding number of cutoffs and thresholds, we agree there are many associated with the filtering (5 total), however 4 of those are fixed for all datasets/tasks and as such do not provide a challenge for automation/future application. We hope this addresses your concerns and please let us know if you have any remaining questions.
>
> **#2 LF-CBM performance without sparsity**
>
> Following your suggestion, we have performed an experiment on LF-CBM performance without the sparsity constraint in Appendix B.6. We can see no sparseness boosts accuracy quite a bit as seen in Table 5, up to 5% on Places365 bringing performance close to that of standard models, but at the same time the models are no longer interpretable as seen by the explanation in Figure 18.
>
> **#3 Comparison between ConceptNet and ImageNet concepts**
>
> Thank you for the suggestion! Following the suggestion, we have conducted an experiment in section B.2 to evaluate how important the use of GPT-3 is to our performance, by comparing against the performance when generating the initial concept sets with ConceptNet. Note that the ConceptNet is not a language model but instead a knowledge graph. The results show that our proposed pipeline with the new concept set from ConceptNet can still give good results in CIFAR, ImageNet and Place365, but does have small decreases in the accuracy (drop ~1%) compared to the GPT-3 concepts. Interestingly, we observed that ConceptNet completely fails on CUB200, while GPT-3 concepts can give good results. We think the above results suggest that our pipeline is effective and the failure of ConceptNet on CUB200 further highlights the importance of our proposed GPT-3 concept filtering.
>
> While we ran out of time to provide qualitative examples, on the difference between the two, what we refer to by higher quality is for example ConceptNet concepts seem quite noisy and often contain typos etc., while GPT-3 concepts are almost always grammatically correct. We would also like to point out that the initial concept set generated by ConceptNet also varies from one task to another as the dataset is constructed by finding concepts with certain relations to the output classes, and we think creating a concept set that works well for all tasks would be extremely challenging.
>
> **#4 Difference in use of CLIP vs Post-hoc CBM**
>
> There is indeed a difference between how we and P-CBM(CLIP) use CLIP. In P-CBM(CLIP), CLIP has to be the backbone model $f(x)$. In comparison, in our model we can use any model as the backbone, and we only need access to CLIP during the training phase to learn the projection layer $W_c$. This gives us a few benefits and flexibility:
> First, we do not need access to CLIP at inference time, which can help a lot in compute/memory constrained settings, while P-CBM(CLIP) must always run the CLIP model during inference.
> Second, we can use a backbone model different from CLIP, such as the specific ResNet-18 trained on CUB-200 which gives better performance on this task requiring specific knowledge while having lower computational cost. However, this is not possible for P-CBM(CLIP) as described above, which is the reason why it is “N/A” for CUB on our Table 2.
>
> **#5 Clarifications**
>
> *In Table 2 you show Label-Free CBM outperforms Post-hoc CBM in 3 of 5 and 2 of 5 tasks for the CLIP variant . You can show values for other datasets if you have them, but without that you can’t say your work clearly outperforms PCBMs “on all datasets”*
>
> Thank you for pointing this out, we have changed this section in 4.1 to “Our LF-CBM noticeably outperforms Post-hoc CBM on the datasets evaluated, but some rows for P-CBM are N/A as they do not provide results for those datasets and it is unclear how to scale their method to larger datasets.”

---

> ### Author Response · Authors · 2022-11-19
> **Author response 2/2**
>
> **#5 Clarifications** continued
>
> *Section 5 starts, “this is the first example of manually editing a large neural network with no obvious flaws”*.
>
> We apologize for the confusion, what we mean by this is that the neural network of previous studies had obvious flaws, not that the evaluation itself had obvious flaws. We think the model editing experiment in P-CBM is an interesting and useful one, but it was done on a model that intentionally had learned a large spurious correlation. In contrast the network we edit does not have any clear flaws. We have changed the sentence to “editing a large well-trained neural network” to avoid confusion.
>
> *How are the decision rules in Fig 3 learned?*
>
> Thank you for pointing out, we did not explain this very well in the original manuscript. We have expanded section 4.3 to say: “The visualization is a Sankey diagram of the final layer weights for two output classes, with the weight between a concept and output class displayed as the width of the line connecting them. We have only included weights with absolute value greater than 0.05 (for comparison the largest weights are usually 0.5-1). Negative weights are denoted as NOT {concept}. “
> So this is simply a visualization of the final layers weights corresponding to two specific output classes, no averaging or other complicated procedures are involved. Let us know if you have further questions.
>
> **#6 Section 5.2 algorithm is vague**
>
> We agree that our algorithm in section 5.2 is more of a proof of concept than a systematic way of doing things and still requires some skill and manual effort, but we think it's an interesting direction for future refinement. We would like to point out that compared to randomly flipping weights our approach lets us identify weights that should be changed with very few tries and gives the editor an intuitive idea about the effects the edit will have. Randomly flipping weights will most likely to decrease the model accuracy as the originally learned weights have already been trained to be close to optimal.
>
> **#7 Nitpicks**
>
> We think it is important to discuss IDBs to have an accurate representation of related works as they are effectively a type of CBM proposed early.
> We appreciate the suggestion but choose to not include models with non-interpretable components for now as they’re not directly comparable (because our goal is to build a fully interpretable model, thus we are comparing only fully interpretable models as our baselines) and including them would cause some issues with clarity and page limit.
> Thank you for the suggestions, we have edited fig 2 accordingly.
>
> **#8 Reproducibility**
>
> The reviewer is correct about the stochastic nature of language models. Therefore, we have experimented with recreating the concept set from GPT-3 and retraining an ImageNet model in section B.3, and found that the accuracy is very similar (71.89% vs average of 71.95% with original concept set) despite the randomness. Note that the specific concepts used and weights learned can be pretty different though all relevant, as shown in Figure 12. To aid with reproducibility and avoid users having to pay for utilizing GPT-3, we will release all our concept sets when we release the code, which we will do prior to publication as stated in section 7 of the original submission.
>
> **#9 New experiments and changes made in this rebuttal period**
>
> We have made many clarifications to the manuscript and performed many new experiments, please see our General Response: Overview of new results and changes in a separate post for a summary of all these changes.
>
> **#10 Summary**
>
> In summary we have:
> - #1 Performed a large ablation study on the effects of concept set filtering, showing our model is not very sensitive to changes in these filters but also showcased the interpretability benefits of using them
> - #2 Performed a study of how our LF-CBM performs without sparsity constraints, showcasing noticeable gains in accuracy but losing most of interpretability
> - #3 Performed an experiment showing how important using GPT-3 is to our overall model performance by comparing against concepts generated by ConceptNet
> - #4 Clarified the difference between how we use CLIP and how P-CBM uses CLIP
> - #5 Clarified several points and edited relevant sections in the manuscript
> - #6 Discussed further our procedure for editing final layer weights
> - #7 Edited Figure 2 to be more informative
> - #8 Performed an experiment on how consistent the concept set generation is with GPT-3 and addressed concerns regarding reproducibility.
> - #9 Discussed our other new experiments and changes made to the manuscript during the rebuttal period
>
> We believe that we have addressed all your concerns. Please let us know if you still have any reservations and we would be happy to address them!

---

> ### Author Response · Authors · 2022-11-29
> **Request for comments**
>
> Thanks again for the review! We have conducted many new experiments (Appendix B) and improved the writing of the manuscript (Please see blue color in the PDF), and we believe we have addressed your concerns. However we have not yet heard from you yet, please let us know what you think of our response and if there are issues remaining, we would be happy to discuss further.

---

> > ### Comment · Reviewer_mHS9 · 2022-11-30
> > **comments on author revisions**
> >
> > Thanks for the extended comments and additional experiments.
> > Although I still feel Section 5.2 and the model editing technique is vague and not very useful as such, that the authors are providing their obtained concept sets for each task is a helpful resource for the community to compare against in future work.  For #2, I do wonder if not using sparsity during learning ( which gives an accuracy boost, but hurts intrpretability ) could simply be applied at the end ( ie, only accept concepts above a threshhold or maximum number per example as being relevant to get the best of both worlds?   For 3, it would still seem having a common concept set provided over many tasks could be beneficial in the future ( do we really need to reinvent concept sets for all task ), I appreciate the comparison.  Thanks for clarifying #4 as well.  In lieu of the changes I've bumped up my evaluation by a point as I still think section 5.2 and a few other things could be improved upon ( though there is a lot of value in the ideas in the paper ).

---

> > > ### Author Response · Authors · 2022-12-01
> > > **Thank you**
> > >
> > > Thank you for the updated review and additional suggestions, we are happy you find the submission improved!
> > > Thanks for your additional suggestions in #2 and #3!
> > >
> > > Regarding 5.2, we would like to emphasize it is one use case of our method and not our main contribution.
> > >
> > > For #2, in general, the accuracy of a sparse model is better if the model is trained to be sparse instead of constraining the model to be sparse after training. For example, see Figure 3 in [R1], which shows that a model trained to be sparse can reach 69.78% accuracy using only top 10 weights per class, but if we restrict a standard trained model to only use top 10 weights per class after the fact its accuracy drops down to 58.46%.
> > >
> > > [R1] Wong, Eric, Shibani Santurkar, and Aleksander Madry. "Leveraging sparse linear layers for debuggable deep networks." International Conference on Machine Learning . PMLR, 2021. (https://arxiv.org/abs/2105.04857)
> > >
> > > For #3, we agree that common concept sets could be useful if the tasks are sufficiently similar. Another useful approach to reduce the need for generating new concepts could be to have a shared base concept set, to which we would add a smaller set of task dependent concepts. This may be an interesting direction to look into in the future.
> > >
> > > Thanks again for your suggestions and please let us know if you have any other questions!

---

> > > ### Author Response · Authors · 2022-12-06
> > > **Clarification on Section 5.2**
> > >
> > > Dear Reviewer mHS9,
> > >
> > > According to your suggestion, we have provided below an improved explanation of Sec 5.2 on how we can edit the final layer weights of our proposed label-free CBM to correct incorrect predictions. Our procedure includes 3 steps:
> > >
> > >
> > > **Step 1: Find an input image $x_i$ where the model makes a Type 4 error (incorrect final layer weight):**
> > >
> > > This is done by visualizing incorrect model predictions and their explanations similar to Figure 4. To identify if it is a Type 4 error, first we check that the ground truth label of the input image is correct and unambiguous (therefore not type 1 error described in sec 5.1). Next, we check that the highly activating concepts look correct (therefore not type 2 or type 3 error described in sec 5.1).
> > >
> > >
> > > **Step 2: Identify a concept to edit:**
> > >
> > > We start by listing the highest activating concepts for the chosen input, i.e. the highest elements of $f_c(x_i)$ (after normalization). This allows us to identify concepts that are the most important for this specific image while not too important for other images. From these concepts we use our domain knowledge (and/or internet search) to understand which concepts are relevant to making this decision.
> > >
> > > For example, the image in the right panel of Figure 6 is originally predicted as a “Shopping basket”, while the ground truth is a “Hamper”. The 5 most highly activating concepts for this image are: “a basket”, “made of rope or string”, “a rope”, “a laundry basket” and “a fishing net”. After a short investigation we find that although the classes “Hamper” and “Shopping basket” are very similar, hampers are more often constructed in a woven like manner. Therefore, we identify the concept of “made of rope or string” as relevant for capturing this difference, and choose it for editing in the next step.
> > >
> > > **Step 3: Change weights by sufficient magnitude:**
> > >
> > > Once we select the concept to be corrected in Step 2, in this step we will compute how much the magnitude of the associated final layer weight should be changed (denoted as $\Delta w$, $\Delta w \in R$) and then edit the weights in the following way:
> > > $W_{F[gt,concept]} \leftarrow W_{F[gt,concept]}+∆w$ and $W_{F[pred,concept]}←W_{F[pred,concept]}−∆w$.
> > >
> > > Note that $W_{F[i,j]}$ denotes the $(i,j)$ element in the matrix $W_F$. The goal of editing is to correct the inaccurate prediction on this specific instance while minimizing effect on other predictions. To calculate $∆w$ needed to flip prediction, we first calculate the difference in logits (before softmax) before the edit, i.e.  $∆a = W_{F[pred,:]} f_c(x_i) - W_{F[gt,:]} f_c(x_i)$.
> > >
> > > Since we would like the prediction to be corrected to gt class, our goal is to design $\Delta w$ such that $(W_{F[gt,:]}+\Delta w \cdot e) f_c(x_i) - (W_{F[pred,:]}−\Delta w \cdot e) f_c(x_i) > 0$, where $e$ is a one-hot row vector with the entry $e_{[concept]} = 1$. Let $b = (W_{F[gt,:]}+\Delta w \cdot e) f_c(x_i) - (W_{F[pred,:]}− \Delta w \cdot e) f_c(x_i)$, where $b$ is a nonnegative constant deciding how large of a margin we want the correct prediction to have. Since $∆a = W_{F[pred,:]} f_c(x_i) - W_{F[gt,:]} f_c(x_i)$, we have: $2 \Delta w \cdot f_c(x_i)_{[concept]}- \Delta a = b$
> > >
> > > $ \Rightarrow \Delta w =  (\Delta a+b)/(2 f_c(x_i)_{[concept]})$
> > >
> > > Typically we use values for $b$ between 0.2 and 2 to calculate required $\Delta w$.
> > >
> > >
> > >
> > > It is worth noting that when the model weights are edited, it might affect other image’s predictions too due to the change of weight parameters. Thus, with the goal to correct the wrong predictions while not affecting other already correct predictions, we suggest each edit should be checked with a validation dataset before applying on a target model.
> > > As described in the end of Sec 5 in the manuscript, with the above proposed model weight editing, we are able to improve the overall model accuracy from 71.98% to 72.02% by performing the model editing for only 5 different images, which is a non-negligible improvement. Since the edits only affect weights for 10/1000 classes, this corresponds to a 4% accuracy boost on the affected classes.
> > >
> > > **Summary:**
> > >
> > > We believe the above detailed explanations have addressed your concerns on the clarity of Sec 5.2, and we will include the above details in the revised manuscript to improve the clarity. Please let us know if you still find the procedure vague or have any reservations, and we would be happy to address them!

---

### Official Review · Reviewer_FHRn · 2022-10-25

**Confidence:** 3
**Correctness:** 3
**Technical Novelty And Significance:** 2
**Empirical Novelty And Significance:** 2
**Recommendation:** 6

**Clarity, Quality, Novelty And Reproducibility:**

I had issues understanding how CLIP-dissect is effectively used during training.

**Strength And Weaknesses:**

Strength: the method provides a framework to build concept bottleneck models that do not require domain experts and labeled concept data.
Weakness: The evaluation is rather weak. The downsides or the limitations of using a model to create concepts are not discussed.

**Summary Of The Paper:**

The paper presents a framework to transform a neural network model into an interpretable CBM model that does not require a domain expert to provide concepts and does not require labeled concept data. The authors leveraged GPT3 model by prompting it in a certain way to get concepts related to each output class. They later perform a set of heuristics to clean the concept list. In order to train a concept bottleneck layer and to learn a projected layer, authors exploited the CLIP-dissect. Finally, they introduced a sparse final layer in order to keep it more interpretable by using the ElasticNet regularization in the loss function. The experiments are conducted using 5 different datasets. The results showed that their method is able to maintain decent performance. Later in the paper, the authors perform preliminary experiments on changing the model's prediction by manipulating the neurons.

**Summary Of The Review:**

- The idea of using GPT3 is smart. I am wondering how the effectiveness of the proposed approach changes, with the change in the model, used to extract concepts.
- The concept filtering is a bit ad-hoc. e.g remove concepts we can't project accurately, remove concepts too similar to each other, concept length longer than 30 characters. I am wondering how much noisy were the initial concepts and how much careful cleaning is required.
- Authors mentioned that the number of concepts is related to the number of classes. Does it mean that it is good to have a one-to-one mapping between concepts and classes? Restricting the neurons to learn a single concept extracted using GPT3, would it not affect the generalization and robustness of the model?
- The number of concepts mentioned in section 4.1. Are these the remaining concepts after cleaning or some further restriction is applied on the number of concept to consider based on the number of classes?
- The evaluation can be improved. The paper focuses a lot on qualitative evaluation. The main quantitative result is on maintaining the overall performance of the model. Authors may add a correlation between the selected concepts and the predicted class. They may also consider ablating concepts and present their effect on the output class.
- Figure 4, I am wondering if authors found false negatives and false positive cases. If a correct concept is ranked the lowest for a certain prediction, this raises the question of the faithfulness of the interpretation.

After rebuttal:
Thank you for the detailed reply and for making substantial improvements to the paper.
- It would be great to add a point on how did you select the number of concepts for a dataset e.g. how did you come up with the number 128 for CIFAR10?

---

> ### Author Response · Authors · 2022-11-19
> **Author response 1/3**
>
> Thank you for the thoughtful review and good suggestions! Below we discuss some concerns you had and how we have addressed them.
>
> **#1 Limitations of using a model to create the concept set**
>
> Thank you for pointing this out, and we have added a Limitations section in Appendix A.1 to discuss the limitations of automated concept set generations and other limitations of our model.
>
> We would like to highlight that, although using a model like GPT-3 to produce concepts is stochastic, and sometimes may fail to generate important concepts for detecting a certain class, our model is quite robust to small changes in the concept set as shown in sections Appendix B.1 and Appendix B.3. I.e. Our model performs consistently well despite these changes. Nevertheless, we think that in general, automatic ways of generating concept sets may sometimes lack the required domain knowledge to include some important concepts, and would be best used in collaboration with a human expert. This would be an interesting future direction to pursue.
>
> **#2 Clarification on the use of CLIP-Dissect**
>
> In essence, CLIP-Dissect is a method to calculate how well the activation pattern $q_k$ of a given neuron k correlates with the presence of a specific concept $t_i$. This is done by calculating how similar the activation pattern of this neuron is to the activation pattern of the CLIP model when prompted with the given concept $P_{:, i}$. This is measured by a specifically designed similarity function, which usually involves normalization or other ways to make comparing two vectors with a very different range of values meaningful. In Equation (1), we propose a well performing similarity function ($\textrm{sim}(q_i,t_i)$) that is fully differentiable, because we would like to backpropagate through it to optimize $W_c$ such that it makes $q_k$ become more similar to $P_{:, k}$, and thus making the neurons activate highly on images containing concept $t_k$. This utilization and modification of CLIP-Dissect is critical to make learning the projection layer possible. Due to space constraints, we weren’t able to include a discussion in the manuscript, but we have fixed typos and clarified section 3.2 to make it more understandable. Hopefully this clarifies the procedure, please let us know if you have any more questions.
>
> **#3 Effect of changing the model to create concept set**
>
> Thank you for the suggestion! Following your suggestion, we have conducted an experiment in section B.2 to evaluate how important the use of GPT-3 is to our performance, by comparing against the performance when generating the initial concept sets with ConceptNet. Note that the ConceptNet is not a language model but instead a knowledge graph. The results show that our proposed pipeline with the new concept set from ConceptNet can still give good results in CIFAR, ImageNet and Place365, but does have small decreases in the accuracy (drop ~1%) compared to the GPT-3 concepts. Interestingly, we observed that ConceptNet completely fails on CUB200, while GPT-3 concepts can give good results. We think the above results suggest that our pipeline is effective and the failure of ConceptNet on CUB200 further highlight the importance of our proposed GPT-3 concept filtering.
>
> **#4 Concept set filtering is ad-hoc**
>
> Thank you for pointing this out! Following your suggestions, we have conducted a large ablation study on the effect of our concept filters, and discussed them and the reasoning behind them further in Appendix B.1. We can see that the performance of our networks in terms of accuracy is not sensitive to the filters used at all, but they are important in improving the interpretability and computational efficiency of our models. In Figure 11 we can see that without using filters, our predictions are still somewhat reasonable but may suffer from issues such as explaining the prediction by the class itself, i.e. *this image is and automobile because it is a car*, and having duplicate concepts such as *car* and a *a car* which makes explanations confusing. Please see our Appendix B.1 for full details and results, including a walkthrough of how the concepts are used and which concepts are removed by each filter for our CIFAR-10 model. We hope this addresses your concerns and please let us know if you have any remaining questions.

---

> ### Author Response · Authors · 2022-11-19
> **Author response 2/3**
>
> **#5 Relationship between number of concepts and number of classes**
>
> *Authors mentioned that the number of concepts is related to the number of classes. Does it mean that it is good to have a one-to-one mapping between concepts and classes?*
>
> We mentioned that the number of classes depends on the number of concepts, however we do not mean that it should be, or that a one-to-one mapping would be desirable. What we meant is that the number of concepts is roughly proportional to the number of output classes. This is mostly an artifact from how we generate the initial concept set, as GPT-3 is prompted with each individual class name and then we take the union of these concepts. This is also somewhat desirable because usually when the number of classes is increased in a machine learning task, it will require more diverse concepts. For example, ImageNet has 1000 classes while cifar10 which only covers 10 classes. Thus, it is intuitive that more concepts are needed to support the decision making of an ImageNet model than a Cifar10 model. For example, it is unlikely that the concepts in Cifar 10 would be sufficient to explain the class “cash machine” in the imagenet, as Cifar10 does not have a class that’s related or close to the class “cash machine”.
>
> In general, we would not expect there to be a one-to-one mapping between the concepts and classes and it would not be desirable because there may be some concepts that can contribute to multiple classes (e.g. a concept “blue” may contribute to classes like “sky”, “bird”, “water” etc). Besides, usually a class also requires multiple concepts (e.g. a class “bird” may relate to “blue”, “feather”, “tail”). This can be clearly seen in the Fig 3 of our original manuscript, where the above two situations could happen at the same time: concept “citrus fruit” has a large weight on both the class “orange” and “lemon”, while the class “orange” has multiple related concepts like “bright orange color”, “a peel” and “citrus fruit”.  In practice we usually need more concepts than we have classes.
>
> We have also changed the wording in section 4.1 to “The number of concepts each model uses is roughly proportional to the number of output classes for that task, as each class adds more initial concepts“ to make the nature of this relationship more clear.
>
> **Number of concepts:**
>
> The remaining concepts in sec 4.1 are the concepts after filtering following the procedures we described in section 3.1.B. There is no further restriction, as like we explained above there is no need to restrict them based on the number of classes.
>
> **#6 Effects on robustness and generalization of enforcing concepts on neurons**
>
> Thanks for your question, this is an interesting point. We note that the concepts are enforced on the neurons in the “concept bottleneck layer” (see Fig 2, step 3 in green-shaded block), which is the second-to-last layer instead of the final layer.
> Generally, it may be preferable to restrict one neuron to one single concept, and to have disentangled concepts for different neurons is essential for making the model interpretable. Ideally, if the concept bottleneck layers contains all the required concepts to explain the classes, the generalization should not be affected. However, if the concept bottleneck layer does not contain all the required concepts, then the accuracy may drop significantly. In fact, this point is demonstrated in our Table 2, where the accuracy of our Label-free CBM is very similar to that of a standard model, while P-CBM has a much larger drop on the accuracy. This showcase our concepts in the concept bottleneck layer are of higher quality than those of P-CBM. We are not sure how this affects the robustness of the model, it could also plausibly make the model more robust but we think that is an interesting direction for future research.
>
> **#7 Correlation between concepts and predicted class**
>
> Thank you for the suggestion, we believe this would be an interesting experiment to run. Unfortunately we were requested to do many experiments by the reviewers and did not have enough time and resources for this experiment by Nov 18. We believe that the newly added 7 experiments in Appendix B (please also see General response) have greatly improved the evaluations. We would be happy to provide additional results after the period Nov 18 if you think this is an important evaluation.

---

> ### Author Response · Authors · 2022-11-19
> **Author response 3/3**
>
> **#8 Faithfulness of explanation**
>
> *If a correct concept is ranked the lowest for a certain prediction, this raises the question of the faithfulness of the interpretation.*  We would like to point out that our explanations are faithful by definition, i.e. they simply report the model weights and neuron activations which are interpretable in the model without the need to learn a local surrogate/approximated model as is done by methods such as LIME or SHAP, therefore we think faithfulness should not be a concern of concept bottleneck models. If the decisions look weird, it means that the model itself is using a weird and hard to understand decision rule or it is poor at estimating concepts, not that our explanation is incorrect. To address your concerns regarding false negatives and false positives, we have included 12 more randomly selected explanations in Appendix B5 Figures 15-17, and we think the explanations are quite good for all of them.
>
> **#9 Weakness of evaluation**
>
> During the rebuttal, we have conducted 7 new experiments discussed in *General response: Overview of new results and changes*. We believe the newly added experiments have greatly improved our evaluation methods. In all evaluations, our results are consistently good and reasonable. Please see the general response for a summary of all our additional results, and changes made to clarify our writing. We hope this addresses your concerns.
>
> **#10 Summary**
>
> In summary we have:
> - #1 Added a discussion section on the limitations of using a model to generate concept set
> - #2 Clarified how we use CLIP-Dissect during training
> - #3 Performed an experiment showing how important using GPT-3 is to our overall model performance by comparing against concepts generated by ConceptNet
> - #4 Performed a large ablation study on the effects of concept set filtering, showing our model is not very sensitive to changes in these filters but also showcased the interpretability benefits of using them
> - #5 Clarified the relationship between number of concepts and number of classes
> - #6 Discussed the effect of creating a concept bottleneck on the generalization/robustness of the models
> - #7 Discussed the proposed experiment on measuring concept effects more
> - #8 Discussed the faithfulness of our explanations and provided more examples of the model decisions on random inputs
> - #9 Discussed all the additional changes and experiments done during the rebuttal period
>
> We believe that we have addressed all your concerns. Please let us know if you still have any reservations and we would be happy to address them!

---

> ### Author Response · Authors · 2022-11-29
> **Request for comments**
>
> Thanks again for the review! We have conducted many new experiments (Appendix B) and improved the writing of the manuscript (Please see blue color in the PDF), and we believe we have addressed your concerns. However we have not yet heard from you yet, please let us know what you think of our response and if there are issues remaining, we would be happy to discuss further.

---

> ### Author Response · Authors · 2022-12-01
> **Thank you**
>
> Thanks for your response, we are glad you find the paper has improved!
>
> Regarding the number of concepts for a dataset (e.g. 128 for CIFAR-10), this was not a number we selected directly but instead a number resulting from our concept set generation and filtering pipeline described in sections 3.1, 3.2 and Appendix B.1. It depends on the initial concept set generation as well as the thresholds used in concept filtering, which were chosen in part to keep the number of concepts is not too large to cause heavy computational overhead but large enough to cover most important concept and retain high accuracy.  We will make this point clearer in the revision, thanks for your comments!

---

### Official Review · Reviewer_CRuQ · 2022-10-28

**Confidence:** 3
**Correctness:** 3
**Technical Novelty And Significance:** 2
**Empirical Novelty And Significance:** 3
**Recommendation:** 8

**Clarity, Quality, Novelty And Reproducibility:**

Overall very interesting new idea. I am not an expert using GPT or CLIP but if the idea of creating concept data is new then I consider this to be an interesting paper !

In terms of clarity, I fail to understand what is happening due to I believe notational issues which hopefully can be clarified during the rebuttal.

In terms of reproducibility, I would be curious to see if there has been any cherry-picking done to obtain the results. The authors seem to only show very few examples and hence I would like the authors to confirm that this in fact works in general

**Strength And Weaknesses:**

Strengths:
- I have not seen people utilize GPT and CLIP in this manner to construct concept datasets (not expert in this field though)
- The idea to create your label-free concept bottleneck paper is very interesting and a step forward to interpretable ML models
- the paper is mostly well and clearly written.
- Given that they use spare matrices they are also able to investigate the model on a sample level as well as edit the model accordingly. i.e. only changing a few weights in the sparse layers.
- This opens up a new avenue for interpretable Deep models.

Weakness:
- I might be wrong but there are some typos in the paper that make it hard for me to understand what steps 2 and 3 are doing.
 - in 3.2 you have $W_c \in d \times N$ shouldn't that we M given figure 2? if f(x) \in R^d what is the dimension for f_c(x)? and therefore what is q_k? I am very confused with the notation.
 - I am pretty sure eq(1) cannot be right. the index of t only goes from 1-M and the index for q (I don't understand). Should this be M?
 - EXPERIMENTS: Please add the 2 standard deviations in the table of results and how many times the algorithm has been run (I might have missed this detail. please point me to the part in the paper)
- EXPERIMENTS: Could the authors please clarify what "Standard (sparse)" means?
- EXPERIMENTS: I would like to investigate the code and see if the authors have cherry-picked any of the examples in the paper. Could the authors please confirm that the examples were not cherry-picked?

**Summary Of The Paper:**

This paper introduces a way to create a concept bottleneck paper without having to provide the concept datasets (which is usually user-specified) (this can be seen as a bug or a feature in the community). The way they achieve this is to first use GPT-3 to query related concepts given an output class. Using these GPT-created concepts they then use CLIP embeddings to match the concerted to images in the training data and hence create their concept dataset without labels. Once this dataset is created they can then train sparse layers i.e. sparse matrices using elastic net regularization to obtain their final predictions.

They then show that their method is able to perform well on datasets like imagenet while still being interpretable.

**Summary Of The Review:**

I have mentioned all my concerns as well as thoughts on the paper in the review above.

I am more than happy to increase my score if the authors are able to answer the concerns above.

---

> ### Author Response · Authors · 2022-11-19
> **Author response 1/2**
>
> Thank you for your positive feedback and comments! Please see our response to your questions below.
>
> **#1 Typos and notations in Sec 3.2**
>
> Thank you for pointing this out! You are correct, we apologize for the typos and have corrected them in the updated manuscript. We have fixed below typos and notations to improve clarity of steps 2 and 3 in sec 3.2:
> - We have changed notation of $d$ -> $d_0$
> - As $f(x) \in R^{d_0}$ and $f_c(x) = W_c f(x)$, thus $W_c$ should be a $M \times d_0$ matrix, and $f_c(x) \in R^M$
> - $q_k$ is the activation pattern vector of neuron $k$ in the concept bottleneck layer on input images $x_1$ to $x_N$, which is formally defined as $q_k^\top = [f_{c,k}(x_1), \ldots, f_{c,k}(x_N)]^\top$, thus $q_k$ is a vector in $R^N$
> - For Eq (1), the summation index should be $i = 1$ to $M$ instead of $N$, which is consistent with the index of $t$ going from $1$ to $M$ as you described.
> We have also edited several parts of section 3 to clarify our description, please see the new text in blue.
>
> **#2 Additional result - standard deviations**
>
> Following your suggestion, we have trained our label-free CBM 3 times, and added the standard deviation of the accuracy into Table 2 for all the 5 datasets in the main text (sec 4.1), please see the blue colors in Table 2 for our edits in the rebuttal period. It can be seen that the standard deviations are quite small (0.06% to 0.29%), demonstrating our Label-free CBM results are consistent, while outperforming prior work P-CBM by a large margin in accuracy: 14.71%-25.9% for P-CBM and 1.9%-9.13% for P-CBM (CLIP). This reiterates the capability of our proposed method to obtain a CBM that is interpretable while not trading-off the accuracy too much.
>
> **#3 Clarification on standard sparse models**
>
> Thank you for pointing this out, while we discussed this in section 4.2, we have increased the clarity in the updated submission changing from: “The standard sparse models were finetuned by us to also have a sparse decision layer with 25-35 non-zero weights per class and have very similar accuracy as our CBM.” -> “The standard sparse models were finetuned by us by learning a sparse final layer directly after feature layer f(x) as described in (Wong et al. 2021) and also have 25-35 nonzero weights per class.”
>
> So effectively these are standard models with the final layer finetuned using the same ElasticNet sparsity objective, hopefully this is more clear.

---

> ### Author Response · Authors · 2022-11-19
> **Author response 2/2**
>
> **#4 Additional results to address cherry-picked concerns**
>
> Thank you for your comments! The examples in Figures 3 and 4 were manually chosen to be informative, but to address your concerns regarding ‘cherry-picking’, we present many additional results similar to Fig 3 and Fig 4 in the original manuscript, that were chosen purely at random in the Appendix B4 and Appendix B5. Specifically:
> - In Appendix B4, we visualize the final layer weights for 4 randomly chosen output classes for both Place 365 and ImageNet.
> - In Appendix B5, we display the decision explanations for 4 randomly chosen input images for 3 datasets: Place365, ImageNet and CUB-200 respectively.
>
> These randomly chosen results showcase that our models are still very explainable on the average case.
>
> **(a) Weight visualization for random classes(Appendix B5)**
> The results are shown in Figures  13 and 14. These results look a little different from Figure 3 as finding two relevant classes to compare requires manual effort and cannot be done at random, so we instead visualize only one class at a time. Note that in Figure 3 we manually colored the lines from concepts according to semantically meaningful groups, which was not done for Figures 13 and 14.
>
> It can be seen that the concepts that have large weights (i.e. thicker lines) are indeed quite relevant to the classes and seem intuitive. For example, in Fig 13, it can be seen that:
> - the class “bathroom” has large weights on the concepts like “a room with a sink”, “a shower curtain”, “bathroom fixture”, “toiletries”;
> - the class “auditorium” has large weights on the concepts like “a speaker’s podium”, “an audience”, “seats for an audience”, “theater”;
> - the class “ice cream parlor” has large weights on the concepts like “a bar or kiosk”, “a serving counter”, “a sundae”, “ice cream”;
> the class “drugstore” has large weights on the concepts like “a pharmacist”, “a pharmacy counter”, “lotions”, “stores”.
>
> Similarly, in Fig 14, it can be seen that:
> - the class “agama” has large weights on the concepts like “a lizard”, “a red, pink or purple color”, “a vibrant blue color”, “a whiptail lizard”;
> - the class “newt” has large weights on the concepts like “bright orange color”, “newts”, “salamander”;
> - the class “marimba” has large weights on the concepts like “a drum-like body”, “a piano bench”, “instrumentalists”;
> - the class “knee pad” has large weights on the concepts like “a knee”, “pads”, “protective gear”.
>
> **(b) Explaining decisions for random images(Appendix B6)**
>
> Overall, it can be seen that the concepts and explanations are of good quality in Figure 15-17, even on the CUB-200 dataset which may require very specific knowledge and concepts to create CBMs. This result demonstrates the effectiveness of our Label-free CBM, which does not require expert-created concept sets unlike existing works on CBMs. For example the Places model detecting “Jacuzzi, indoor“ using concepts like “a large, round tub” and “a room designated for bathing” in Figure 15 which seem like good concepts to use for such a decision.
> To sum up, we think that the additional results for multiple random examples are all pretty good and reasonable, and we hope this eliminates your concerns.
>
> **#5 Code release**
>
> We are committed to releasing our code prior to publication.
>
> **#6 New experiments and changes made in this rebuttal period**
>
> We have made many clarifications to the manuscript and performed many new experiments, please see our General Response: Overview of new results and changes in a separate post for a summary of all these changes.
>
> **#7 Summary**
>
> In summary we have:
>
> - #1 Fixed the typos and clarified description in section 3
> - #2 Trained all models 3 times and reported standard deviations, showing our results are consistent
> - #3 Clarified our description of training the standard sparse models.
> - #4 Generated many additional random explanations of model weights and individual decisions in Appendices B.4 and B5 to show our results are good overall and not caused by cherry-picking
> - #5 Confirmed that we will release the code prior to publication
> - #6 Discussed the many other experiments and changes to the manuscript we have performed
>
> Please feel free to let us know if you have any additional questions or comments, and we would be happy to discuss further!

---

> > ### Comment · Reviewer_CRuQ · 2022-12-11
> > **Response**
> >
> > First of all sorry for the late reply as I was away for a personal matter and then fell sick right after ...
> >
> > I appreciate the new changes and will raise my score accordingly!
> >
> >
> > Best

---

> > > ### Author Response · Authors · 2022-12-11
> > > **Thank you**
> > >
> > > Thank you for the response and positive comments! Hope you're feeling better.

---

> ### Author Response · Authors · 2022-11-29
> **Request for comments**
>
> Thanks again for the review! We have conducted many new experiments (Appendix B) and improved the writing of the manuscript (Please see blue color in the PDF), and we believe we have addressed your concerns. However we have not yet heard from you yet, please let us know what you think of our response and if there are issues remaining, we would be happy to discuss further.

---

> ### Author Response · Authors · 2022-12-05
> **Request for comments (reminder)**
>
> Dear Reviewer,
>
> As the review period is approaching its end, we would like to follow up on our request for comments. We believe we have addressed all your concerns regarding experiments, clarity of explanation and reproducibility and would like to hear your thoughts in response to our rebuttal. Please let us know if you still have any reservations and we would be happy to address them. Thanks for your time!

---

### Author Response · Authors · 2022-11-19
**General response: Overview of new results and changes**

Thank you for all the constructive feedback and suggestions. In response to the reviews, we have conducted many new experiments, mostly focused on ablating the importance of different parts of our proposed method and providing more results to showcase the capability of our method as requested by the reviewers.

**New experiments:**

We have compiled all the new results and discussion in the Appendix B (p.16-p.26) in the PDF and marked the new text and edits in blue. Please see below short summary of all the new experiments that we have done in this rebuttal period:


1. We have conducted an ablation study and additional discussion on the effect of our concept set filtering on the results in Appendix B.1. The result shows that our models have consistently good performance and are not very sensitive to specific filter choices, while also highlighting the usefulness of our proposed filters in sec 3.1 to improve interpretability.

2. We added an ablation study in Appendix B.2 on using ConceptNet to generate the initial concept set instead of our proposed GPT-3 method in sec 3.1. The results show that using the ConceptNet as suggested by reviewesr generally reduces accuracy slightly for common benchmarks (CIFAR, ImageNet and Place365) and does not work at all for CUB200, which shows that our proposed GPT-3 method is a better choice to generate concepts.

3. We tested regenerating the initial concept set and how much it changes our results in Appendix B.3. The results show that despite the concept set changes due to randomness of GPT-3, the resulting Label-Free CBMs have very little change in accuracy, showing the consistency of our method.

4. We added additional visualizations of final layer weights similar to Figure 3 for randomly chosen output classes in Appendix B.4, and it can be seen that the results are all quite good and reasonable

5. We added additional explanations of decisions similar to Figure 4 for randomly chosen inputs on 3 different datasets in Appendix B.5, and it can be seen that the results are all quite good and reasonable

6. We evaluated the accuracy and interpretability of our LF-CBM trained without sparsity constraint on the final layer weights in Appendix B.6. We see it significantly boosts accuracy but greatly hurts interpretability of the model, showing the need of our proposed sparse final layer.

7. We trained all our models 3 times and edited Table 2 to report the mean and standard deviation of our training runs, showcasing that our results are consistently good.


**Changes:**

- Added section Appendix A.1 to discuss the limitations of our model, such as automated concept set generation and applicability on tasks requiring specific knowledge.
- Corrected typos with M vs N in equations of section 3.2
- Clarified that the final number of concepts is reduced after final filtering step in section 3.2 and 3.3
- Corrected typos/inclarity in section 3.3
- Changed some notation such as $W_{F0}$ -> $b_F$ and $d$ -> $d_0$
- Added dimension of parameters to improve clarity
- Clarified the connection between number of classes and number of concepts in section 4.1
- Clarified training procedure of standard (sparse) models in section 4.2.
- Changed outperforms PostHoc-CBM on all datasets -> on datasets evaluated and modified wording in section 4.2
- Explained how our visualization in section 4.3 was created.
- In Section 5, changed “a large neural network with no obvious flaws” -> “a large well-trained neural network” to avoid confusion
- Edited figure 2 According to reviewer suggestions to improve clarity
- Several other clarifications and corrections of typos.

---

### Author Response · Authors · 2023-06-05
**Update: Crowdsourced evaluation of interpretability**

We conducted a crowdsourced evaluation on how interpretable our models are. The results are available in the Appendix B on arxiv: https://arxiv.org/abs/2304.06129

---

### Decision · Program_Chairs · 2023-01-20

**Decision:**

Accept: poster

**Justification For Why Not Higher Score:**

From my perspective this is an interesting interpretability paper - it seems to innovate over prior work and also includes a nice study of manual editing of a classifier. The specific focus (concept bottleneck models) is pretty niche and the method itself is pretty complicated, so I think it may have limited impact. But it's worth publishing.

**Justification For Why Not Lower Score:**

All reviewers recommended acceptance.

**Metareview: Summary, Strengths And Weaknesses:**

This paper proposes a new way to learn a "concept bottleneck model", which maps activations in the penultimate layer of a classifer to human-understandable concepts. Compared to prior methods for learning concept bottleneck models, the proposed method does not require hand-specifying the concepts and also retains strong accuracy even on large-scale tasks. The paper includes experiments on a few vision datasets, but also includes an interesting study on manually editing the classification layer in a model to improve its performance based on insights gained from the concept bottleneck model. While reviewers had various suggestions for improved clarity in the initial review, all reviewers ultimately recommended acceptance.

**Note From Pc:**

if the above contains the word "oral" or "spotlight" please see: "oral" presentation means -> notable-top-5% and "spotlight" means -> notable-top-25%. As stated in our emails, we are disassociating presentation type from AC recommendations